# MOMEMTO: Patch-based Memory Gate Model in Time Series Foundation Model

## Abstract

Recently reconstruction-based deep models have been widely used for time series anomaly detection, but as their capacity and representation capability increase, these models tend to over-generalize, often reconstructing unseen anomalies accurately. Prior works have attempted to mitigate this by incorporating a memory architecture that stores prototypes of normal patterns. Nevertheless, these approaches suffer from high training costs and have yet to be effectively integrated with time series foundation models (TFMs). To address these challenges, we propose **MOMEMTO**, an improved variant of TFM for anomaly detection, enhanced with a patch-based memory module to mitigate over-generalization. The memory module is designed to capture representative normal patterns from multiple domains and enables a single model to be jointly fine-tuned across multiple datasets through a multi-domain training strategy. MOMEMTO initializes memory items with latent representations from a pre-trained encoder, organizes them into patch-level units, and updates them via an attention mechanism. We evaluate our method using 23 univariate benchmark datasets. Experimental results demonstrate that MOMEMTO, as a single model, achieves higher scores on AUC and VUS metrics compared to baseline methods, and further enhances the performance of its backbone TFM, particularly in few-shot learning scenarios.

## 1 Introduction

Complex cyber-physical systems operate continuously, accumulating vast volumes of sequential time series data from multiple sensors. Monitoring these systems and detecting anomalies at an early stage is highly beneficial in terms of operational cost. In real-world anomaly detection scenarios, it is common to simultaneously handle multi-domain datasets that originate from different application areas (e.g., distinct sensor types or process configurations). Such scenarios frequently involve data imbalance and limited labeled data. We formulate this problem as an unsupervised learning task to tackle these challenges.

Reconstruction-based deep models are widely used for unsupervised time series anomaly detection. In this approach, an encoder-decoder network is trained with a self-supervised objective to reconstruct input sequences. These models aim to reproduce normal patterns precisely, while generating higher reconstruction errors for anomalies. Various models have been proposed within this paradigm, including OmniAnomaly (Su et al., 2019), Anomaly Transformer (Xu et al., 2022), TranAD (Tuli et al., 2022), and TimesNet (Wu et al., 2023). Recent models have achieved strong performance due to the representation learning capability of neural networks. However, existing reconstruction-based approaches often suffer from over-generalization, where anomalous inputs are reconstructed with high accuracy (Gong et al., 2019; Park et al., 2020). This issue arises when the encoder captures distinctive features of anomalies or when the decoder has sufficient capacity to reconstruct abnormal representations.

Recent research has increasingly focused on TFMs, which are pre-trained on large-scale time series data from a variety of sources. TFMs can be adapted to new tasks with minimal additional training through approaches such as multi-task learning, few-shot learning, and prompting. However, applying TFMs to anomaly detection presents two main challenges. First, most TFMs, such as MOIRAI (Woo et al., 2024), TimesFM (Das et al., 2024), Chronos (Ansari et al., 2024), Time-MoE (Shi et al., 2024), and Sundial (Liu et al., 2025) are primarily designed for time series forecasting. They often

employ decoder-only architectures that limit their suitability for detection tasks. Second, TFMs used for anomaly detection typically adopt reconstruction-based approaches, leaving them vulnerable to over-generalization.

To address these challenges, we adapt the Gate memory module of MEMTO (Song et al., 2023), a reconstruction-based deep model for time series anomaly detection. This module stores memory items that represent the prototypical features of normal patterns. When anomalies are reconstructed using these normal memory items, the resulting outputs resemble normal samples, thus alleviating over-generalization. However, MEMTO is highly sensitive to memory initialization, requiring encoder pre-training to obtain informative and stable representations. It also follows a one-model-per-dataset framework, which leads to high training costs when handling multiple datasets. In addition, MEMTO uses a point-level memory rather than a patch-based structure, which reduces its effectiveness in detecting interval or periodic anomalies (Shen, 2025).

Furthermore, we adopt the pre-trained encoder from MOMENT (Goswami et al., 2024), a TFM trained on large-scale time series datasets. MOMENT employs a patch-level masked representation learning strategy, where input sequences are divided into fixed-length patches and trained to reconstruct the masked patches. Its encoder, built on the T5 architecture (Raffel et al., 2020), can be coupled with task-specific decoders, providing highly transferable time series representations and flexibility across diverse tasks. However, like other TFMs, MOMENT adopts multi-domain pre-training, but achieving competitive performance typically requires fine-tuning a separate model for each target dataset. This limitation prevents it from fully exploiting the benefits of multi-domain representations. Moreover, as a reconstruction-based model for anomaly detection, MOMENT still remains vulnerable to over-generalization.

Building upon previous methods, we introduce MOMEMTO, an improved variant of MOMENT adapted for reconstruction-based time series anomaly detection. The model integrates the pre-trained encoder from MOMENT with the patch-based memory module. Unlike MEMTO, our model benefits from MOMENT's encoder, which already possesses sufficient representational capacity. This provides well-initialized features for the memory module and alleviates MEMTO's sensitivity to memory initialization. The patch-based memory module stores encoder outputs that are organized at the patch level. Whereas the original memory items stored prototypical features of normal patterns restricted to a single dataset, the patch-based memory items are designed to capture representative normal patterns from multiple domains. This design alleviates over-generalization while enabling the model to learn effectively in diverse domains. Specifically, multi-domain training refers to jointly fine-tuning a single model across datasets from diverse application areas, rather than training a separate model for each dataset. This strategy facilitates knowledge sharing across domains and allows more effective use of limited data, while also reducing computational cost in both training time and memory usage. Under the multi-domain training strategy, a single MOMEMTO model achieves strong results on 23 univariate benchmark datasets compared to baseline methods. In few-shot learning scenarios, it demonstrates significant improvements over its backbone TFM. Our contributions can be summarized as follows:

- We introduce a patch-based memory architecture to support TFMs, which stores features from multi-domain datasets and is broadly applicable to various patch-based methods.

- MOMEMTO extends MOMENT into a TFM specialized for time series anomaly detection, supporting multi-domain training and leveraging the patch-based memory module to mitigate over-generalization.

- MOMEMTO achieves strong results on 23 univariate benchmark datasets and demonstrates superior few-shot anomaly detection performance.

## 2 RELATED WORK

**Transformers and patching for time series** Transformers (Vaswani et al., 2017) have demonstrated strong performance on sequential data processing. For time series analysis, Transformers benefit from the self-attention mechanism and are used to capture reliable long-range temporal dependencies (Kitaev et al., 2020; Li et al., 2021; Zhou et al., 2021; Liu et al., 2024). Among them, PatchTST (Nie et al., 2023) is a Transformer-based model specifically designed for time series anal-

ysis. Its first strategy is patching, in which input sequences are segmented into subsequence-level patches, preserving local semantic information while reducing computational complexity. Its second strategy is a channel-independent processing scheme, where each univariate series is processed independently with shared Transformer weights, minimizing inter-channel interference and effectively capturing distinct temporal dynamics. Together, these designs enhance both the efficiency and the representation capability of Transformers for time series tasks.

**Memory-guided deep models** Recently, memory architectures have been introduced to enhance neural models by enabling external storage and retrieval of long-term information. They have been applied in diverse domains such as natural language processing, including large-scale language modeling (Lample et al., 2019), retrieval-augmented generation (Lewis et al., 2020), and long-context understanding (Wang et al., 2023), and in computer vision tasks like video captioning (Lei et al., 2020), video object segmentation (Oh et al., 2019) as well as in reinforcement learning for episodic memory and sample-efficient decision making (Le et al., 2021). For anomaly detection, several approaches in computer vision employ memory modules to store features of normal patterns (Zhou et al., 2023). MemAE (Gong et al., 2019) integrates a memory module into an autoencoder. MNAD (Park et al., 2020) introduces a weighted memory-update strategy to capture temporal anomalies in videos. Based on these concepts, MEMTO is the first model to introduce a memory architecture into time series anomaly detection, employing a data-driven update mechanism. More recently, H-PAD (Shen, 2025) extends this idea by learning hybrid patch- and period-level prototypes to capture both interval and periodic anomalies.

MOMEMTO extends MEMTO by refining its memory mechanism to explicitly capture patch-level patterns. Our patch-based memory module is designed based on patching and channel-independent strategies, allowing selective updates of memory items rather than updating all items as in MEMTO. This design enables the memory module to learn more informative features of normal patterns while also improving the computational efficiency of memory updates compared to the original memory module (Nie et al., 2023; Song et al., 2023).

## 3 METHOD

We consider multiple univariate time series datasets, which are categorized into distinct domains. Each domain exhibits unique temporal characteristics such as length, sampling frequency, trend, seasonality, and noise. A single univariate time series from a dataset $X$ is denoted as $x$. When divided into non-overlapping patches of fixed length $L$, the sequence is represented as $x \in \mathbb{R}^{P \times L}$, where $P$ is the number of observed patches. If the number of observed patches $P$ is smaller than the maximum number of patches $N$ that the model can process, we apply zero-padding so that the sequence matches the shape $\mathbb{R}^{N \times L}$.

### 3.1 MOMEMTO

Figure 1 illustrates the overall architecture of MOMEMTO, which mainly consists of an encoder–decoder with the patch-based memory module. The input time series $x$ is first processed by the encoder. The encoder output representation $q \in \mathbb{R}^{P \times d_{model}}$ is used as queries to interact with the memory module, where $d_{model}$ denotes the embedding dimension. The query vectors are compared with the stored memory items to retrieve the most relevant ones, and the selected memory items are subsequently updated conditioned on the queries. The queries are then refined via memory-based attention. The updated queries $\tilde{q} \in \mathbb{R}^{P \times d_{model}}$ are combined with the original queries $q$ (i.e., $[q; \tilde{q}] \in \mathbb{R}^{P \times 2d_{model}}$) and passed to the decoder. The decoder maps the combined query representation to the input space to reconstruct the time series $\hat{x} \in \mathbb{R}^{P \times L}$.

#### 3.1.1 PRE-TRAINED ENCODER AND DECODER

We adopt the pre-trained encoder from MOMENT-large as the backbone for our model. The encoder receives a univariate time series $x \in \mathbb{R}^{N \times L}$ together with a mask vector of length $N$, where each entry is 0 or 1 to indicate unobserved and observed patches, respectively. Each input patch is treated as a token and embedded into a vector of length $d_{model}$. Patch-level self-attention is then applied to capture temporal dependencies and generate contextualized representations. For reconstruction, we

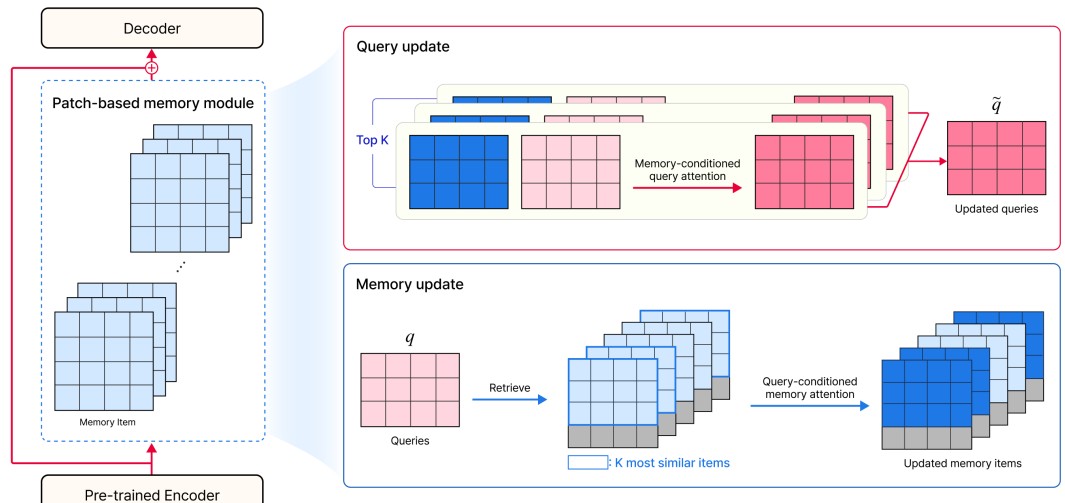

Figure 1: Architecture of MOMEMTO.

use a shared lightweight decoder implemented as two fully-connected layers. This design mitigates the risk of over-generalization and promotes reliance on the encoder's representational capacity, while enabling the model to generalize across datasets with varying sequence lengths.

### 3.1.2 PATCH-BASED MEMORY MODULE

To alleviate over-generalization in reconstruction-based models, we introduce the patch-based memory module as a redesigned variant of MEMTO's original memory module. It naturally aligns with the encoder's patch representations. The module consists of $M$ memory items $\{m_i\}_{i=1}^{M}$, where each memory item $m_i \in \mathbb{R}^{N \times d_{model}}$ stores a prototypical normal pattern at the patch level. The module operates in a data-driven manner through five stages: memory initialization, memory alignment, memory update, query update, and query alignment. Through these stages, the memory items and queries are progressively refined and aligned with representative normal patterns.

**Memory initialization**   Each memory item $m_i$ is initialized with a specific domain, which may encompass multiple datasets. It is stored as the mean of encoder representations from $S_i$ instances randomly sampled across the datasets belonging to that domain. These encoder representations are denoted by $\{q_s\}_{s=1}^{S_i}$. The initial value of $m_i$ is then computed as

$$m_i = \frac{1}{S_i} \sum_{s=1}^{S_i} q_s, \quad i = 1, \ldots, M.$$

To ensure stable updates, memory items are consistently L2-normalized at the patch level.

**Memory alignment**   After initializing the memory items, we first identify the observed patches in the encoder output $q$ using the input mask and retrieve the corresponding subset of representations. To ensure consistent interaction, each memory item $m_i$ is aligned with the same subset of observed patches, and the queries are likewise L2-normalized, denoted as $\hat{q}$.

$$
\begin{array}{ccc}
q \in \mathbb{R}^{N \times d_{model}} & & q \in \mathbb{R}^{P \times d_{model}} \\
m_i \in \mathbb{R}^{N \times d_{model}} & \xrightarrow{\text{retrieve}} & m_i \in \mathbb{R}^{P \times d_{model}}
\end{array}
$$

This alignment process allows the model to handle time series of varying lengths flexibly and selectively update memory items, enhancing adaptability across datasets.

**Memory update** We then compute the similarity between the normalized queries $\hat{q}$ and the corresponding memory slices using dot products. Instead of updating every memory item, we select the top-$K$ memory items with the highest similarity scores. These selected memory items $m_{1st}, m_{2nd}, \ldots, m_{K\text{-}th}$ are updated through query-based patch-level attention, where $m_{i\text{-}th}$ denotes the memory item with the $i$-th highest similarity to $\hat{q}$. The resulting updates are proportionally applied to the selected memory items in a data-driven way, with the remaining update procedure following MEMTO's standard strategy (details in Appendix B.4).

**Query update** In the query update stage, we refine $\hat{q}$ into an updated representation $\tilde{q}$ using memory-based patch-level attention. This process is performed only on the top-$K$ memory items selected in the previous stage. For each selected memory item $m_{i\text{-}th}$, we compute an intermediate representation $\tilde{q}_{i\text{-}th}$. The final refined representation $\tilde{q}$ is obtained by taking the weighted sum of the $K$ intermediate representations:

$$\tilde{q} = w_{1st} \cdot \tilde{q}_{1st} + w_{2nd} \cdot \tilde{q}_{2nd} + \cdots + w_{K\text{-}th} \cdot \tilde{q}_{K\text{-}th}$$

where $w_{i\text{-}th}$ denotes the similarity score between $\hat{q}$ and $m_{i\text{-}th}$, computed in the previous stage.

**Query alignment** The resulting $\tilde{q}$ is then combined with the original $q$ in a patch-aligned manner, where each patch-level memory item contributes only to the corresponding patch position. This combined representation is then used as the input to the shared decoder. This ordered memory structure preserves the positional structure of the sequence and captures the underlying patterns in each patch.

### 3.2 Multi-domain Training

Unlike the conventional one-model-per-dataset paradigm, we train a single unified model that is jointly optimized on the entire dataset collection, enabling it to detect anomalies in time series with heterogeneous lengths and characteristics. During multi-domain training, the patch-based memory module is updated in a data-driven manner, allowing memory items to adapt to representative normal patterns from different domains. At initialization, the number of memory items is set equal to the number of user-defined domains, which provides a balanced starting point but does not enforce a strict one-to-one correspondence. Through data-driven updates, a single memory item may accumulate information from multiple domains and evolve into a domain-general feature. This training strategy facilitates knowledge sharing while maintaining a single model architecture in which the memory captures diverse normal patterns.

## 4 Experiments

**Datasets** We evaluate our model on the TSB-AD-U benchmark (Liu & Paparrizos, 2024), consisting of 870 univariate time series across 23 datasets. The benchmark is constructed through an organized curation process: both univariate and multivariate datasets are collected, with multivariate data converted into univariate channels. Channels unrelated to anomaly labels, such as categorical, binary, or noisy channels, are discarded. Mislabeled or low-quality series are removed through algorithmic tests and human review, and balanced sampling is applied to ensure fairness. The collection also includes sequences from public benchmarks such as MSL (Mars Science Laboratory rover), SMD (Server Machine Dataset), and the UCR Anomaly Archive (Wu & Keogh, 2022). Further dataset details are provided in Appendix A.

**Implementation details** We generate subsequences by applying a non-overlapping window with a length of 512 to obtain fixed-length inputs for each time series. We evaluate anomaly detection performance using threshold-independent metrics, such as the Area Under Curve (AUC) and Volume Under the Surface (VUS) (Paparrizos et al., 2022). More detailed information on hyperparameter settings can be found in Appendix B.

## 4.1 MAIN RESULTS

In this experiment, we evaluate the performance of MOMEMTO on time series anomaly detection by comparing it with 13 baseline methods. The baselines consist of three categories: classic approaches (OCSVM (Schölkopf et al., 1999), LOF (Breunig et al., 2000), Isolation Forest (Liu et al., 2008)), Deep learning models (LSTM-AD (Malhotra et al., 2015), Donut (Xu et al., 2018), OmniAnomaly (Su et al., 2019), TranAD (Tuli et al., 2022), Anomaly Transformer (Xu et al., 2022), MEMTO, TimesNet (Wu et al., 2023), KAN-AD (Zhou et al., 2025), DADA (Shentu et al., 2025)), and pretrained TFMs (TimesFM, Chronos, MOMENT).

Table 1: Model performance comparison on the benchmark. Metrics are reported as percentages (%). Subscripts indicate training settings: $md$ = multi-domain joint fine-tuning of a single model across all datasets; models without a subscript denote the one-model-per-dataset setting.

| Category | Model | AUC-PR | AUC-ROC | VUS-PR | VUS-ROC |
|---|---|---|---|---|---|
| Classical | LOF | 14.24 | 58.17 | 16.87 | 67.74 |
| | OCSVM | 15.85 | 65.13 | 22.51 | 73.49 |
| | Isolation Forest | 29.04 | 71.18 | 29.80 | 77.51 |
| Deep learning | MEMTO | 6.90 | 52.05 | 11.30 | 56.30 |
| | Donut | 15.28 | 55.97 | 20.78 | 68.56 |
| | TimesNet | 18.10 | 61.19 | 25.91 | 72.03 |
| | Anomaly Transformer | 19.02 | 62.53 | 23.61 | 73.05 |
| | TranAD | 24.67 | 63.66 | 29.95 | 73.87 |
| | KAN-AD | 29.20 | 69.54 | 30.25 | 76.30 |
| | OmniAnomaly | 29.87 | 69.45 | 32.15 | 76.49 |
| | LSTM-AD | 31.77 | 67.70 | 32.66 | 75.74 |
| | DADA | 33.59 | 70.42 | 34.06 | 77.62 |
| Foundation | Chronos | 26.43 | 65.64 | 27.18 | 72.91 |
| | TimesFM | 28.43 | 67.29 | 29.88 | 74.20 |
| | MOMENT | 29.45 | 69.97 | 29.73 | 76.84 |
| | **MOMEMTO** | 32.83 | 70.00 | 33.23 | 77.49 |
| | **MOMEMTO**$_{md}$ | **36.35** | **74.83** | **37.62** | **80.95** |

Table 1 shows the evaluation results on the benchmark. Higher values in these metrics indicate more accurate anomaly detection. Among all compared models, **MOMEMTO**$_{md}$, as a single model, achieves the best performance across all evaluation criteria. Moreover, reconstruction-based TFMs such as MOMENT and MOMEMTO yield stronger results than other TFMs that are primarily designed for forecasting tasks. A direct comparison with its backbone TFM, MOMENT, shows that MOMEMTO consistently surpasses its backbone across metrics and settings. This demonstrates that the patch-based memory module effectively mitigates over-generalization and improves detection capability, even when built upon the same pre-trained encoder, highlighting the importance of architectural design tailored to anomaly detection.

**Few-shot learning** To evaluate the robustness of our approach under limited data scenarios, we conduct a few-shot learning experiment. In this setting, the ratio of training data is gradually varied from 10% to 90%, and the performance of MOMEMTO is compared with MOMENT under the condition that only a small fraction of samples is available for fine-tuning.

Figure 2 shows the effect of training data ratio on AUC-PR. Across all ratios, MOMEMTO consistently outperforms MOMENT, highlighting the advantage of the patch-based memory module. The gap is substantial in the few-shot regime (10%–30%), where both the one-model-per-dataset and multi-domain variants achieve noticeably higher scores than the corresponding MOMENT variants. As the proportion of training data increases, MOMEMTO continues to improve in both settings, while MOMENT shows marginal or no further improvement. Moreover, MOMEMTO$_{md}$ achieves consistently higher performance than MOMEMTO without multi-domain training, indicating that multi-domain training further strengthens the memory module by improving data efficiency. Overall, these results demonstrate that MOMEMTO is robust under limited data and benefits even more from additional data when trained in the multi-domain setting.

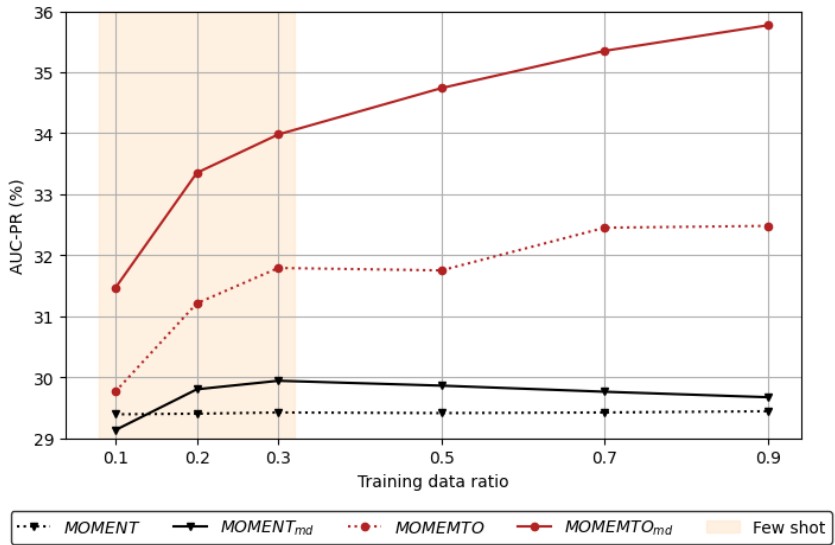

Figure 2: Effect of the training data ratio on AUC-PR.

## 4.2 MODEL ANALYSIS

**Ablation study**  To investigate the contribution of each component in MOMEMTO, we conduct an ablation study by varying the encoder initialization (scratch vs. pre-trained) and the use of the patch-based memory module.

As shown in Table 2, both components contribute significantly to performance improvement. Replacing a scratch encoder with a pre-trained one improves AUC-PR (22.44 → 29.70), while adding the memory module to a scratch encoder yields an even larger gain (22.44 → 33.63). The combination of both components achieves the best performance, reaching an AUC-PR of 36.35. Similar trends are consistently observed across other metrics. These results indicate that the two components provide complementary benefits: the patch-based memory module mitigates over-generalization, and the pre-trained encoder provides a strong representational foundation that amplifies the memory module's effectiveness.

Table 2: Ablation results of pre-trained encoder and patch-based memory module. 'PMM' denotes the patch-based memory module.

| Encoder | PMM | AUC-PR | AUC-ROC | VUS-PR | VUS-ROC |
|---------|-----|--------|---------|--------|---------|
| scratch | × | 22.44 | 67.35 | 18.90 | 66.50 |
| scratch | ○ | 33.63 | 72.17 | 34.65 | 78.78 |
| pre-trained | × | 29.70 | 69.60 | 30.69 | 76.58 |
| pre-trained | ○ | **36.35** | **74.83** | **37.62** | **80.95** |

**Comparison of anomaly score distributions**  Figure 3 presents the distributions of anomaly scores for normal and anomalous samples across a subset of domains. Each boxplot shows the distributions of normal and anomalous samples within a domain, enabling direct comparison of the models' discriminative ability. Additional results for the remaining domains are provided in the Appendix C.5. Overall, MOMENT exhibits limited separation between normal and anomalous samples, with substantial overlap in their score distributions. In contrast, MOMEMTO often yields a clearer margin between the two groups: anomaly scores for anomalous samples tend to shift toward higher values, whereas those of normal samples remain clustered around low values. This tendency

suggests that MOMEMTO improves the distinction between normal and anomalous patterns in the majority of domains.

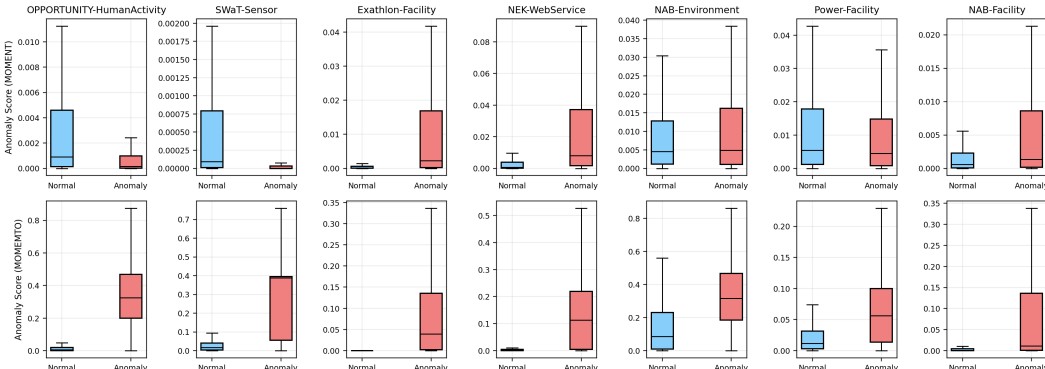

Figure 3: Comparison of anomaly score distributions between normal (blue) and anomalous (red) samples across a subset of domains. The upper row shows the results obtained with the backbone model MOMENT, while the lower row corresponds to our proposed model.

**Data-driven memory utilization** We visualize in Figure 4 how memory items are accessed for subsequences from different domains. Each heatmap displays the accumulated similarity between the true domain of the input and the selected memory domains, normalized such that each row sums to one. The left illustrates updates during the entire training, and the right shows references by new inputs at test time. With the data-driven strategy, no domain labels are provided during training or testing. Updates tend to concentrate on a subset of memory items, rather than being evenly distributed across domains, and these items are consistently referenced at test time. To validate the effectiveness of our strategy, we compare it with two alternative strategies: (i) freezing the memory items without updates and (ii) updating only the memory item initially assigned to the input's domain (own-domain update). Both baselines use the memory item corresponding to the input's domain during training and testing. Table 3 summarizes the performance comparison. Even though the initial allocation of memory items to domains does not remain aligned during training, our data-driven strategy still achieves higher performance while utilizing fewer memory items than alternative strategies. This indicates that the memory module has learned a stable set of domain-general features, which are effectively reused across unseen inputs.

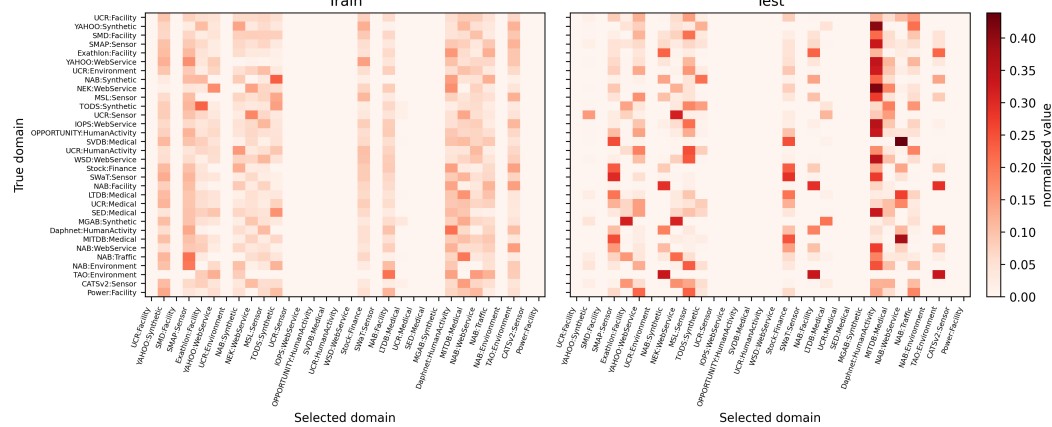

Figure 4: Visualization of row-normalized accumulated domain-to-memory similarity over the entire training process (left) and at test time (right). Each heatmap shows how subsequences from a true domain reference memory items across domains.

Table 3: Performance comparison of different memory update strategies. *No update*: memory frozen after initialization; *Own-domain update*: update only the item initially assigned to the input's domain; *Data-driven update (ours)*: update the similarity-retrieved item regardless of initial domain. At test time, both baselines use the memory item corresponding to the input's domain.

| Strategy | AUC-PR | AUC-ROC | VUS-PR | VUS-ROC |
|---|---|---|---|---|
| No update | 34.97 | 73.77 | 36.91 | 80.19 |
| Own-domain update | 35.16 | 73.64 | 36.60 | 80.06 |
| Data-driven update (ours) | **36.35** | **74.83** | **37.62** | **80.95** |

**Computational efficiency**   We measure the training time, inference time, and model size under different configurations of the patch-based memory module (PMM) and the multi-domain training paradigm, as summarized in Table 4. Here, *context switching* refers to the additional overhead incurred when multiple separately trained models must be loaded and executed sequentially, such as repeated model loading, GPU memory allocation, and parameter initialization. While the inference time remains stable at around 54 seconds across all configurations, the total training time and model size differ drastically. Training separate models introduces excessive context switching overhead (around 1,300 seconds) and substantially enlarges the total model size beyond 450 GB, which is computationally expensive. In contrast, our unified model with PMM achieves efficient total training (87.39 seconds) and maintains a compact size (1.3 GB).

Table 4: Comparison of training, context switching, inference, and model size under different configurations of the patch-based memory module (PMM) and multi-domain training. The configuration without PMM is equivalent to MOMENT.

| | Time (sec) | | | | Model size (GB) |
|---|---|---|---|---|---|
| | Training | Context switching | Total training | Inference | |
| MOMEMTO | 82.22 | 5.17 | 87.39 | 54.61 | 1.30 |
| w/o PMM | 75.83 | 4.59 | 80.42 | 54.23 | 1.29 |
| w/o multi-domain training | 135.77 | 1311.51 | 1447.28 | 54.54 | 455.74 |
| w/o PMM & multi-domain training | 108.96 | 1294.26 | 1403.22 | 54.29 | 451.64 |

## 5 CONCLUSION AND FUTURE WORK

We introduce MOMEMTO, a time series foundation model for anomaly detection enhanced with a patch-based memory module. By leveraging a pre-trained encoder and the patch-based memory module, our approach mitigates over-generalization and achieves superior performance across 23 benchmark datasets, with notable gains in few-shot and multi-domain settings. While MOMEMTO shows promising results, several important aspects remain unexplored. The experiments focus primarily on a univariate benchmark, and the study does not include a theoretical analysis of MOMEMTO. Future work will investigate these aspects and broaden the applicability of our approach.

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

## A TSB-AD-U BENCHMARK

In this section, we provide details of the TSB-AD-U benchmark (Liu & Paparrizos, 2024) used in our experiments. The benchmark comprises 23 univariate time series datasets with a total of 870 time series, collected from diverse domains. Table 5 summarizes the dataset composition, including the number of series, average sequence length, anomaly counts, and anomaly ratios. The benchmark is divided into two partitions: the Eval set, which contains 350 time series and is designated for evaluation, and the Tuning set, which contains 48 time series and is used for hyperparameter optimization. Table 6 summarizes the statistics of the Eval and Tuning sets.

Table 5: Summary characteristics of 23 datasets included in TSB-AD-U. '-' in the 2nd column indicates this dataset is transformed from the multivariate dataset. The 'Category' column indicates whether the datasets feature point anomalies (P) or sequence anomalies (Seq).

| Name | # TS Collected | # TS Curated | Avg Dim | Avg TS Len | Avg # Anomaly | Avg Anomaly Len | Anomaly Ratio | Category |
|------|---------------|--------------|---------|-----------|---------------|-----------------|---------------|----------|
| UCR | 250 | 228 | 1 | 67818.7 | 1 | 198.9 | 0.6% | P&Seq |
| NAB | 58 | 28 | 1 | 5099.7 | 1.6 | 370.1 | 10.6% | Seq |
| YAHOO | 367 | 259 | 1 | 1560.2 | 5.5 | 2.5 | 0.6% | P&Seq |
| IOPS | 58 | 17 | 1 | 72792.3 | 25.6 | 48.7 | 1.3% | Seq |
| MGAB | 10 | 9 | 1 | 97777.8 | 9.7 | 20.0 | 0.2% | Seq |
| WSD | 210 | 111 | 1 | 17444.5 | 5.1 | 25.4 | 0.6% | Seq |
| SED | 6 | 3 | 1 | 23332.3 | 14.7 | 64.0 | 4.1% | Seq |
| TODS | 15 | 15 | 1 | 5000.0 | 97.3 | 18.7 | 6.3% | P&Seq |
| NEK | 48 | 9 | 1 | 1073.0 | 2.9 | 51.1 | 8.0% | P&Seq |
| Stock | 90 | 20 | 1 | 15000.0 | 1246.9 | 1.1 | 9.4% | P&Seq |
| Power | 1 | 1 | 1 | 35040.0 | 4 | 750 | 8.5% | Seq |
| Daphnet (U) | - | 1 | 1 | 38774.0 | 6 | 384.3 | 5.9% | Seq |
| CATSv2 (U) | - | 1 | 1 | 300000.0 | 19.0 | 778.9 | 4.9% | Seq |
| SWaT (U) | - | 1 | 1 | 419919.0 | 27.0 | 1876.0 | 12.1% | Seq |
| LTDB (U) | - | 9 | 1 | 99700.0 | 127.5 | 144.5 | 18.6% | Seq |
| TAO (U) | - | 3 | 1 | 10000.0 | 838.7 | 1.1 | 9.4% | P&Seq |
| Exathlon (U) | - | 32 | 1 | 44075.8 | 3.1 | 1577.3 | 11.0% | Seq |
| MITDB (U) | - | 8 | 1 | 631250.0 | 68.7 | 451.9 | 4.2% | Seq |
| MSL (U) | - | 9 | 1 | 3492.0 | 1.3 | 130.0 | 5.8% | Seq |
| SMAP (U) | - | 19 | 1 | 7700.2 | 1.2 | 210.1 | 2.8% | Seq |
| SMD (U) | - | 38 | 1 | 24207.7 | 2.4 | 173.7 | 2.0% | Seq |
| SVDB (U) | - | 20 | 1 | 171380.0 | 36.4 | 292.5 | 3.6% | Seq |
| OPP (U) | - | 29 | 1 | 16544.8 | 1.4 | 653.4 | 6.4% | Seq |

Table 6: Statistics of the TSB-AD-U benchmark splits.

|  | Split | # TS | Avg Length | Avg Anomaly Length | Avg # Anomalies | Anomaly Ratio |
|--|-------|------|-----------|--------------------|-----------------|---------------|
| TSB-AD-U | All | 870 | 38814.1 | 179.5 | 39.7 | 2.4% |
|  | Eval | 350 | 51886.7 | 321.3 | 46.6 | 4.5% |
|  | Tuning | 48 | 47143.3 | 185.9 | 82.6 | 3.5% |

## B IMPLEMENTATION DETAILS

All experiments in this paper are conducted on the Eval set of the TSB-AD-U benchmark, which contains 350 univariate time series. We describe below how the domain is defined, the training settings we consider, and the training configurations we adopt in our experiments.

### B.1 DOMAIN PARTITION

We assign a domain label to each time series in the Eval set. A domain label is defined as a tuple (Dataset, Sub-domain), where the Dataset indicates the source collection and the Sub-domain specifies the actual domain of the time series. Time series characteristics may vary across domains within the same dataset, and they may also differ across datasets even when they share the same domain. To capture this heterogeneity, domain labels serve as a heuristic for balanced memory initialization.

Each file in the benchmark follows a structured naming convention.

For example:

`001_NAB_id_1_Facility_tr_1007_1st_2014.csv`

In this case:

- Dataset:= `NAB`
- Sub-domain:= `Facility`
- **Domain label**:= `(NAB, Facility)`

In total, the 350 univariate time series in the Eval set are grouped into 32 domains according to this labeling. These domain labels are used only during the memory initialization stage.

## B.2 TRAINING SETTINGS

We distinguish between two training settings used in our experiments:

- **One-model-per-dataset:** In this conventional scheme, a separate model is trained for each dataset in the Eval set. Since the Eval set contains 350 time series, this setting requires 350 independently trained models.
- **Multi-domain training:** In contrast, our proposed approach jointly optimizes a single model across all datasets in the Eval set.

Unless otherwise specified, all baseline models follow the one-model-per-dataset setting, while MO-MEMTO adopts the multi-domain training setting.

## B.3 BASELINES AND HYPERPARAMETER SETTINGS

All baseline models are implemented by training on the train set and predicting the test set as defined in the benchmark. Their hyperparameters are primarily tuned on the tuning set following the benchmark protocol. Most models segment each time series into fixed-length windows using a sliding window with stride 1. A subset of models, such as MOMENT, Anomaly Transformer, and MEMTO, use non-overlapping windows to respect the original implementations in their respective papers.

Table 7: Hyperparameter settings for baseline models.

| Model | Hyperparameter |
|---|---|
| LOF | n_neighbors = 50, metric = minkowski |
| OCSVM | periodicity = 2, kernel = rbf |
| Isolation Forest | n_estimators = 200 |
| MEMTO | win_size = 100, lr = 0.00005, $\lambda$ = 0.01, n_memory = 10 |
| Donut | win_size = 60, lr = 0.0001 |
| TimesNet | win_size = 32, lr = 0.0001 |
| AnomalyTransformer | win_size = 100, lr = 0.001 |
| TranAD | win_size = 10, lr = 0.0001 |
| OmniAnomaly | win_size = 5, lr = 0.002 |
| LSTM-AD | win_size = 100, lr = 0.0008 |
| Chronos | win_size = 100 |
| TimesFM | win_size = 96 |
| MOMENT | win_size = 512, lr = 0.0001 |

For MOMEMTO, we use MOMENT-large (Goswami et al., 2024) as the backbone model, and most of the hyperparameters remain consistent with the original MOMENT configuration. Only the parameters related to the patch-based memory module are newly introduced. Table 8 summarizes the hyperparameter settings and environment used in our experiments.

Table 8: Hyperparameter settings for MOMEMTO and experimental environment.

| Category | Setting |
|---|---|
| Window size | 512 |
| Patch length ($L$) | 8 |
| Patch stride length | 8 |
| Number of patches ($N$) | 64 |
| $d_{model}$ | 1024 |
| Number of referenced items ($K$) | 3 |
| Temperature parameter ($\tau$) | 0.3 |
| Learning rate | 0.0001 |
| Epochs | 2 |
| Optimizer | Adam |
| Loss function | Mean Squared Error (MSE) |
| GPU | NVIDIA GeForce RTX 4090 (24GB) |
| Framework | PyTorch 2.7.0 |

## B.4 ALGORITHMS

Algorithms 1 and 2 illustrate the overall mechanism of our model. They present the matrix operation version of the forward process when a single input subsequence is given.

---

**Algorithm 1** Proposed Method **MOMEMTO**

---

**Input** $x \in \mathbb{R}^{N \times L}$ : input time series, $\texttt{[mask]} = \{0,1\}^N$, $\mathcal{M} \in \mathbb{R}^{M \times N \times d_{model}}$: domain-specific initialized memory items

**Training params** $f_e$: encoder, $f_d$: decoder, $f_m$: patch-based memory module

1: $q \leftarrow f_e(x, \texttt{[mask]})$          $\triangleright\, q \in \mathbb{R}^{P \times d_{model}}$

2: $\hat{q} \leftarrow \text{L2Norm}(q, dim = 1)$

3: $\tilde{q} \leftarrow f_m(\hat{q}, \mathcal{M}, \texttt{[mask]})$

4: $q \leftarrow concat([q, \tilde{q}], dim = 1)$          $\triangleright\, q \in \mathbb{R}^{P \times (2 \cdot d_{model})}$

5: $\hat{x} \leftarrow f_d(q)$

6: **return** $\hat{x}$          $\triangleright$ Reconstructed time series

---

**Algorithm 2** Proposed **Patch-based memory module**

**Input** $\hat{q} \in \mathbb{R}^{P \times d_{model}}$: normalized queries, $\texttt{[mask]} = \{0,1\}^N$, $\mathcal{M} \in \mathbb{R}^{M \times N \times d_{model}}$: memory items

**Training params** $U_\psi, W_\psi \in \mathbb{R}^{d_{model} \times d_{model}}$: linear projection matrices

1: $\mathcal{M} \in \mathbb{R}^{M \times P \times d_{model}} \xleftarrow{\texttt{[mask]}} \mathcal{M} \in \mathbb{R}^{M \times N \times d_{model}}$         ▷ Memory alignment

2: Reshape: $\hat{q} \in \mathbb{R}^{1 \times (P \cdot d_{model})}, \mathcal{M} \in \mathbb{R}^{M \times (P \cdot d_{model})}$

3: $\lambda \leftarrow softmax(\hat{q}\mathcal{M}^T)$

4: $(\lambda_{1st} : m_{1st}), (\lambda_{2nd} : m_{2nd}), ..., (\lambda_{K\text{-}th} : m_{K\text{-}th}) \leftarrow TopK(\lambda)$    ▷ Find K most similar items

5: Reshape: $\hat{q} \in \mathbb{R}^{P \times d_{model}}, \mathcal{M} \in \mathbb{R}^{M \times P \times d_{model}}$

6: $\tilde{q} \leftarrow \mathbf{0}_{P \times d_{model}}$

7: **for** $i \leftarrow 1$ **to** $K$ **do**                                            ▷ Memory update

8:      $v \leftarrow softmax(m_{i\text{-}th}q^T)$          ▷ $m_{i\text{-}th} \in \mathbb{R}^{P \times d_{model}}$: the $i\text{-}th$ similar item

9:      $\psi \leftarrow sigmoid(m_{i\text{-}th}U_\psi + vqW_\psi)$        ▷ $\psi \in \mathbb{R}^{P \times d_{model}}$: update rate

10:      $\tilde{m}_{i\text{-}th} \leftarrow (1 - \psi) \odot m_{i\text{-}th} + \psi \odot vq$

11:      $w \leftarrow softmax(q(\tilde{m}_{i\text{-}th})^T)$               ▷ Query update

12:      $\tilde{q}_{i\text{-}th} \leftarrow w\tilde{m}_{i\text{-}th}$

13:      $\tilde{q} \leftarrow \tilde{q} + (\lambda_{i\text{-}th} \odot \tilde{q}_{i\text{-}th})$

14: **end for**

15: **return** $\tilde{q}$                                                       ▷ $\tilde{q} \in \mathbb{R}^{P \times d_{model}}$

# C ADDITIONAL EXPERIMENTAL RESULTS

## C.1 ADDITIONAL RESULTS FOR BENCHMARK ACCURACY EVALUATION

Table 9 summarizes more detailed VUS-PR results by domain, aggregated from 350 time series across 32 domains.

Table 9: VUS-PR results by domain.

| Domain | LOF | OCSVM | IForest | MEMTO | Donut | TimesNet | A.T. | TranAD | OmniAnomaly | LSTM-AD | Chronos | TimesFM | MOMENT | MOMEMTO | MOMEMTO$_{md}$ |
|---|---|---|---|---|---|---|---|---|---|---|---|---|---|---|---|
| (CATSv2, Sensor) | 6.38 | 26.08 | 7.94 | 5.31 | 8.09 | 7.23 | 9.34 | 7.51 | 10.92 | **29.16** | 9.60 | 24.50 | 19.81 | 9.79 | 25.45 |
| (Daphnet, HumanActivity) | 13.06 | 6.09 | 36.43 | 6.40 | 9.69 | **40.19** | 12.85 | 21.05 | 27.42 | 13.27 | 31.53 | 35.46 | 34.95 | 20.55 | 11.42 |
| (Exathlon, Facility) | 19.85 | 29.05 | 66.72 | 13.46 | 51.21 | 53.96 | 64.43 | 78.09 | 86.74 | 73.28 | 42.24 | 50.19 | 52.79 | 79.88 | **90.48** |
| (IOPS, WebService) | 12.05 | 6.91 | **27.71** | 4.57 | 10.07 | 20.95 | 10.94 | 19.98 | 23.55 | 22.81 | 19.16 | 18.72 | 21.28 | 25.90 | 22.05 |
| (LTDB, Medical) | 26.23 | 32.71 | 33.79 | 24.72 | 30.85 | 31.69 | **39.20** | 31.43 | 31.62 | 34.66 | 26.11 | 27.60 | 29.37 | 32.14 | 36.97 |
| (MGAB, Synthetic) | 0.44 | 0.73 | 0.45 | 0.43 | 0.42 | 0.43 | 0.42 | 0.49 | 0.49 | **4.47** | 0.45 | 0.44 | 0.51 | 0.44 | 1.25 |
| (MITDB, Medical) | 6.12 | 14.09 | 9.79 | 6.17 | 9.81 | 7.89 | 8.26 | 9.34 | 9.52 | 12.15 | 5.88 | 6.46 | 7.73 | 9.74 | **17.13** |
| (MSL, Sensor) | 14.70 | 28.26 | 28.83 | 11.19 | 22.18 | **32.25** | 27.33 | 25.57 | 28.08 | 24.88 | 22.41 | 31.05 | 32.19 | 25.32 | 29.00 |
| (NAB, Environment) | 25.50 | 7.74 | 35.90 | 10.54 | 11.33 | 12.38 | 33.82 | 32.45 | 31.96 | 20.20 | 10.92 | 10.92 | 13.21 | 37.32 | **38.20** |
| (NAB, Facility) | 15.95 | 20.87 | 23.89 | 12.06 | 16.79 | 21.59 | 26.51 | 23.79 | 25.93 | 18.93 | 18.73 | 20.17 | 25.20 | 24.18 | **30.85** |
| (NAB, Synthetic) | 18.23 | **52.23** | 27.27 | 15.67 | 18.72 | 23.30 | 21.14 | 23.49 | 24.55 | 23.40 | 20.05 | 18.96 | 23.62 | 20.96 | 31.77 |
| (NAB, Traffic) | 14.06 | **20.66** | 16.66 | 11.46 | 16.67 | 16.04 | 14.40 | 17.82 | 19.90 | 11.68 | 15.43 | 15.82 | 16.91 | 16.76 | 17.71 |
| (NAB, WebService) | 16.34 | 17.25 | 16.53 | 16.03 | 17.76 | **27.59** | 20.13 | 16.05 | 19.06 | 16.15 | 19.79 | 16.28 | 19.21 | 15.98 | 24.31 |
| (NEK, WebService) | 37.70 | 25.66 | 59.30 | 8.98 | 48.00 | 48.48 | 30.15 | 70.90 | **85.24** | 74.16 | 33.84 | 34.46 | 42.57 | 73.24 | 66.39 |
| (OPPORTUNITY, HumanActivity) | 14.25 | 11.49 | 43.39 | 16.58 | 28.21 | 5.25 | 67.89 | 65.17 | 69.39 | 58.43 | 5.36 | 4.80 | 6.06 | 74.48 | **77.91** |
| (Power, Facility) | 9.05 | 16.36 | 8.20 | 9.05 | 8.56 | 7.55 | 9.27 | 14.46 | 14.83 | 6.73 | 8.18 | 7.89 | 8.22 | 10.53 | 23.59 |
| (SED, Medical) | 10.98 | 5.98 | 35.84 | 7.66 | 10.53 | 4.90 | 6.31 | 4.65 | 5.23 | 5.72 | 6.22 | 5.28 | 5.85 | 5.93 | 7.90 |
| (SMAP, Sensor) | 15.12 | 51.18 | 24.66 | 5.96 | 30.43 | 42.65 | 15.91 | 24.29 | 25.36 | 25.02 | 17.66 | 33.02 | 35.09 | 21.14 | 26.38 |
| (SMD, Facility) | 13.01 | 8.15 | 34.18 | 3.43 | 28.62 | **53.85** | 14.28 | 34.42 | 38.02 | 50.31 | 34.09 | 39.99 | 38.72 | 34.45 | 33.31 |
| (SVDB, Medical) | 5.26 | **20.47** | 9.03 | 4.75 | 7.80 | 8.59 | 11.96 | 7.99 | 8.86 | 13.38 | 5.51 | 6.31 | 7.14 | 8.42 | 15.91 |
| (SWaT, Sensor) | 12.12 | 9.35 | 49.69 | 12.71 | 46.75 | 9.60 | **67.15** | 51.45 | 46.08 | 66.78 | 20.64 | 17.09 | 7.87 | 46.68 | 47.70 |
| (Stock, Finance) | 74.99 | 73.15 | 99.30 | 76.31 | 78.00 | 78.52 | 74.86 | 82.21 | 92.54 | 85.41 | 98.39 | 98.55 | **99.52** | 98.49 | 88.97 |
| (TAO, Environment) | 90.94 | 92.49 | 98.52 | 90.95 | 90.68 | 90.49 | 91.26 | 94.19 | 97.79 | 99.87 | 99.85 | 99.35 | **99.99** | 99.99 | 97.15 |
| (TODS, Synthetic) | 48.55 | 65.07 | 51.81 | 45.38 | 48.97 | 58.01 | 50.07 | 47.78 | 46.41 | 47.23 | 73.24 | 75.15 | 67.15 | 49.85 | 61.41 |
| (UCR, Environment) | 2.26 | **20.86** | 0.86 | 0.87 | 0.97 | 1.64 | 1.31 | 0.79 | 0.92 | 1.02 | 14.27 | 14.81 | 3.31 | 1.06 | 4.33 |
| (UCR, Facility) | 1.01 | **3.33** | 1.06 | 0.81 | 0.70 | 1.14 | 2.73 | 1.18 | 1.29 | 1.17 | 0.69 | 0.89 | 1.07 | 1.13 | 2.71 |
| (UCR, HumanActivity) | 3.92 | 1.39 | 3.44 | 0.85 | 1.57 | 3.51 | 1.84 | 2.48 | 3.38 | 2.47 | 5.62 | 5.72 | 5.97 | 2.42 | **10.93** |
| (UCR, Medical) | 2.25 | **16.16** | 2.64 | 1.64 | 1.97 | 2.79 | 2.15 | 2.38 | 2.46 | 2.32 | 7.44 | 7.43 | 4.09 | 2.31 | 5.52 |
| (UCR, Sensor) | 2.29 | **53.98** | 4.14 | 1.45 | 1.58 | 1.91 | 5.54 | 2.82 | 2.97 | 5.19 | 3.66 | 3.60 | 5.13 | 2.49 | 4.64 |
| (WSD, WebService) | 8.61 | 2.81 | 14.41 | 4.29 | 5.19 | 19.16 | 4.78 | 12.03 | 15.47 | 13.38 | 17.67 | **20.62** | 20.27 | 15.79 | 17.23 |
| (YAHOO, Synthetic) | 62.91 | 27.06 | 56.39 | 18.36 | 8.47 | 40.14 | 3.13 | 36.24 | 14.70 | 63.67 | 90.57 | 93.75 | **97.34** | 63.57 | 82.97 |
| (YAHOO, WebService) | 17.23 | 19.71 | 33.72 | 12.39 | 7.48 | 13.09 | 11.63 | 24.28 | 31.59 | 30.89 | 70.28 | **72.16** | 63.98 | 35.62 | 44.04 |

## C.2 ZERO-SHOT ANOMALY DETECTION

We conduct a leave-one-out evaluation across 32 domains to assess the zero-shot performance of MOMEMTO. In each iteration, one domain is held out as the target domain, while the model is trained on the remaining 31 domains. The trained model is then directly applied to the held-out domain without any fine-tuning, under a zero-shot setting. Performance metrics are computed individually for each target domain and subsequently aggregated over all 32 domains. To highlight the effectiveness of our memory-augmented design, we also compare MOMEMTO against its backbone MOMENT.

Table 10: Zero-shot anomaly detection results under leave-one-out evaluation across 32 domains. MOMEMTO is compared with MOMENT.

|  | AUC-PR | AUC-ROC | VUS-PR | VUS-ROC |
|---|---|---|---|---|
| $\text{MOMENT}_{md}$ | 29.68 | 69.61 | 30.64 | 76.57 |
| $\text{MOMEMTO}_{md}$ | **34.51** | **72.17** | **35.76** | **78.32** |

## C.3 NUMBER OF REFERENCED MEMORY ITEMS

We analyze how the number of referenced memory items ($K$) and the training data ratio jointly affect the performance of MOMEMTO. As shown in Figure 5, when $K = 1$, the model leverages only the most similar memory item, providing stable yet limited improvements. For $K \geq 2$, the performance is not always superior to that with $K = 1$, but the best results are consistently achieved when the model references multiple items.

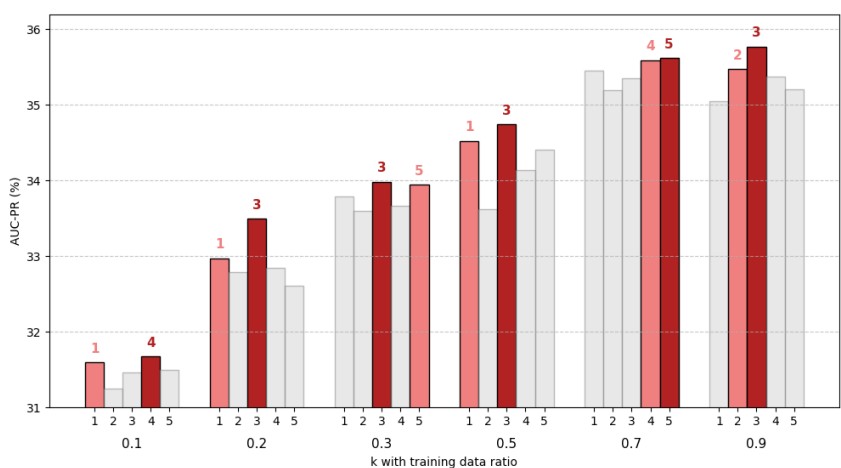

Figure 5: Effect of the number of referenced memory items ($K$) and the training data ratio on AUC-PR.

## C.4 NUMBER OF MEMORY ITEMS

To evaluate how the number of memory items affects MOMEMTO, we conduct a sensitivity analysis by varying the number of memory items from 64 down to 1. In this experiment, each memory item is initialized using K-means centroid values.

Figure 6 visualizes the row-normalized scaled domain-to-memory similarities at the beginning and end of training. Across all configurations, the model consistently relies on a small subset of memory items, regardless of the total number of memory items.

Table 11: Sensitivity analysis of the number of memory items on the main benchmark.

|         | 64    | 32    | 16    | 8     | 4     | 2     | 1     |
|---------|-------|-------|-------|-------|-------|-------|-------|
| AUC-PR  | 36.11 | 36.05 | 36.26 | 36.07 | 35.97 | 35.69 | 35.81 |
| AUC-ROC | 74.59 | 74.22 | 74.18 | 74.19 | 74.30 | 73.25 | 73.50 |

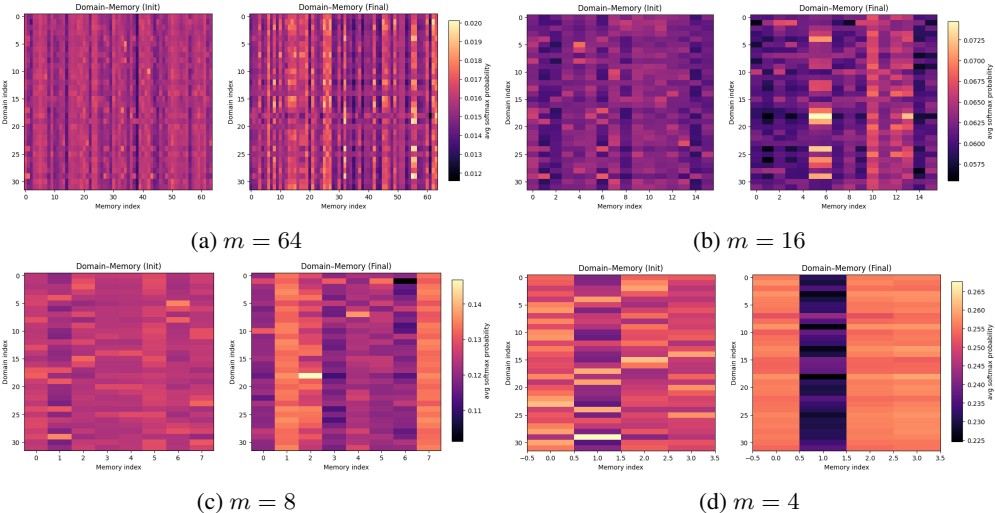

(a) $m = 64$        (b) $m = 16$

(c) $m = 8$        (d) $m = 4$

Figure 6: Scaled domain-to-memory similarity maps at the beginning (left) and end of training (right) for different numbers of memory items. Each heatmap is row-normalized.

Table 12: Few-shot performance sensitivity to the number of memory items (AUC-PR)

|    | 0.1   | 0.2   | 0.3   | 0.5   | 0.7   | 0.9   |
|----|-------|-------|-------|-------|-------|-------|
| 32 | 31.02 | 32.84 | 33.40 | 35.04 | 35.23 | 35.63 |
| 16 | 30.98 | 32.83 | 33.46 | 35.01 | 35.36 | 35.55 |
| 8  | 30.95 | 32.75 | 33.48 | 34.81 | 35.21 | 35.16 |
| 4  | 30.93 | 32.42 | 33.52 | 34.69 | 35.42 | 35.32 |
| 2  | 30.85 | 32.80 | 33.52 | 35.08 | 35.67 | 35.65 |
| 1  | 30.93 | 32.32 | 33.50 | 34.84 | 35.60 | 35.51 |

These results indicate that MOMEMTO exhibits stable performance with respect to the number of memory items, consistent with the sensitivity analysis reported in MEMTO.

## C.5 ANOMALY SCORE DISTRIBUTIONS

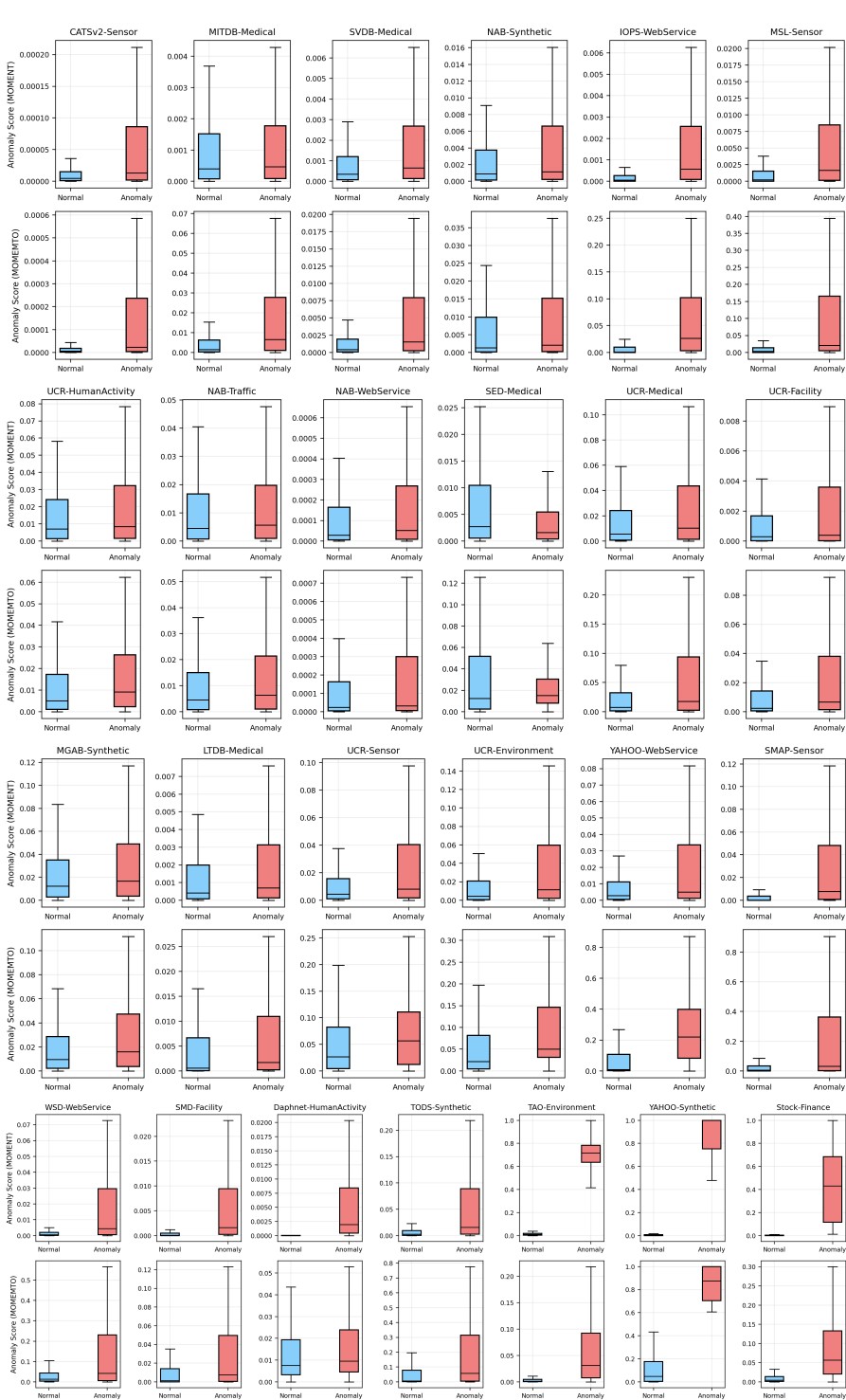

Figure 7: Comparison of anomaly score distributions between normal (blue) and anomalous (red) samples across a subset of domains. The upper row shows the results obtained with the backbone model MOMENT, while the lower row corresponds to our proposed model.

## D  THE USE OF LARGE LANGUAGE MODELS

A large language model (LLM) is used exclusively for linguistic polishing of the manuscript, including grammar correction and style refinement. The LLM does not contribute to the conception of ideas, experimental design, implementation, or analysis.

