# OpenReview forum: "MOMEMTO: Patch-based Memory Gate Model in Time Series Foundation Model"
_ICLR.cc/2026/Conference — Submitted to ICLR 2026_

### Official Review · Reviewer_5wBe · 2025-10-25

**Soundness:** 3
**Presentation:** 2
**Contribution:** 1
**Rating:** 4
**Confidence:** 4

**Summary:**

This paper addresses the over-generalization problem in anomaly detection models by extracting patch memories from different domains and reconstructing time series through linear combinations of these patch memories, thereby improving the accuracy of anomaly detection.

**Strengths:**

1. The problem that this paper aims to solve is clearly described, and the proposed solution is well articulated.

2. The paper demonstrates strong technical soundness, showing good practical and methodological rationality.

3. The authors conduct extensive experiments on 870 time series, comparing their approach against 11 popular time series anomaly detection methods, which convincingly validates the effectiveness of the proposed method.

**Weaknesses:**

1. Compared with MEMTO, the contributions and innovations of this paper are rather marginal. Both approaches build a prototype memory and reconstruct the original data through linear combinations of vectors within that memory. Overall, the work lacks novelty and makes limited contributions to the advancement of the field.

2. Although the paper points out that MEMTO heavily depends on memory initialization, the discussion of how the proposed method alleviates this issue is insufficient. For example, it is unclear why the proposed method is less dependent on memory initialization and whether there are experiments that support and verify this claim.

3. The paper states that MEMTO follows a “one-model-per-dataset” paradigm. However, the original MEMTO paper mentions that MEMTO can capture multiple prototypes to represent diverse patterns, implying that it could generalize across datasets and potentially achieve a “one-model-for-multiple-datasets” capability. Therefore, this claim in the reviewed paper requires further justification.

4. The core problem this paper aims to solve is preventing model over-generalization, which can lead to reduced reconstruction errors for anomalies and thus hinder anomaly discrimination. However, the experiments do not directly demonstrate whether the proposed method effectively mitigates this problem — for instance, by showing that its recall rate improves compared with baselines at their best F1 scores.

**Questions:**

1. Could you provide an additional experiment comparing the recall of the proposed method and the baselines at their best F1 scores?

2. Could you elaborate on how the proposed method addresses MEMTO’s strong dependency on memory initialization?

3. Could you summarize and highlight the specific contributions and innovations of the proposed method compared with MEMTO?

---

> ### Author Response · Authors · 2025-11-21
>
> We sincerely thank the reviewer for the constructive feedback, which greatly helped us clarify our contributions and improve the quality of our work.
> ***
> > **Q1.** Could you provide an additional experiment comparing the recall of the proposed method and the baselines at their best F1 scores?
>
> We report the precision, recall, and F1 scores based on the best-F1 setting in the table below. To further illustrate the effect of mitigating over-generalization, we also include an additional supporting metric, the two-sample Kolmogorov-Smirnov (KS) statistic. The KS statistic measures the maximum difference between the cumulative distributions of normal and anomalous scores, which reflects the degree of separation between the two score distributions. A larger KS value indicates a clearer separation between normal and anomalous scores.
>
>
> | Model         | P     | R      | F1     | KS statistic |
> |--------------|:-----:|:------:|:------:|:------------:|
> | LOF          | 30.10 | 47.15  | 20.50  | 24.21        |
> | OCSVM        | 20.86 | 60.16  | 21.72  | 40.81        |
> | Isolation Forest | 32.42 | $\underline{61.39}$ | 34.95 | 47.39 |
> | MEMTO        | 22.31 |  6.68  |  7.06  |  6.46        |
> | Donut        | 22.54 | 59.65  | 21.28  | 37.15        |
> | TimesNet     | 26.21 | 54.11  | 23.52  | 38.08        |
> | AT           | 25.06 | 57.11  | 23.58  | 33.99        |
> | TranAD       | 36.80 | 59.72  | 30.03  | 41.79        |
> | OmniAnomaly  | 38.65 | **62.03** | 34.32 | 47.14 |
> | LSTM-AD      | $\underline{44.12}$ | 60.40 | 37.09 | $\underline{49.49}$ |
> | Chronos      | 35.92 | 52.38  | 31.78  | 36.40        |
> | TimesFM      | 36.57 | 55.12  | 33.51  | 39.96        |
> | MOMENT       | 38.27 | 50.58  | 33.83  | 42.70        |
> | MOMENTO | 44.07 | 57.02 | $\underline{37.43}$ | 46.91 |
> | MOMENTO$_{md}$   | **45.66** | 57.60 | **40.50** | **53.43** |
>
>
> Several models, including OmniAnomaly, Isolation Forest, and OCSVM, show higher average recall than our proposed model. Meanwhile, MOMEMTO exhibits competitive performance across the other metrics. Furthermore, when comparing our method with the backbone model MOMENT, we observe improvements in both recall and the KS-statistic, suggesting an enhancement in anomaly discrimination capability.
> ***
> >**Q2.** Could you elaborate on how the proposed method addresses MEMTO’s strong dependency on memory initialization?
>
> Thank you for the insightful question. MEMTO exhibits a strong dependency on memory initialization because its memory items are constructed from encoder representations that are not yet stable early in training. When these representations vary across instances, the initialized memory items may not reflect consistent normal patterns, and the subsequent memory updates can amplify this variability. This can lead to unstable optimization or inconsistent anomaly detection performance. For this reason, MEMTO introduces an additional encoder warm-up stage to stabilize the representations before initializing the memory.
>
>
> In our method, this dependency is alleviated by initializing memory items using latent representations from a pretrained MOMENT encoder. Since the encoder is trained on a large-scale corpus, its patch-level representations are more consistent across instances at the beginning of fine-tuning. This provides well-structured starting points for memory initialization without requiring a separate warm-up procedure.
>
>
> We experimentally validate this effect in our ablation study (Table 2 in the paper). We directly compare the memory module when used with a scratch encoder and with the pretrained encoder to assess how different initialization conditions influence performance. In the scratch encoder setting, using the memory module improves AUC PR from 22.44 to 33.63. When the memory module is used together with the pretrained encoder, the score further increases to 36.35. These results indicate that the pretrained encoder provides a more reliable initialization for the memory module and reduces the sensitivity to initialization that is observed in MEMTO.

---

> ### Author Response · Authors · 2025-11-21
>
> ***
> >**Q3.** Could you summarize and highlight the specific contributions and innovations of the proposed method compared with MEMTO?
>
>
> We summarize below the key differences between our method and MEMTO.
>
>
> **(1) Improved memory initialization through the use of a pre-trained encoder**
>
>
> This point was discussed in the previous response.
>
> **(2) From time point memory to patch-level memory: structural advantages for anomaly detection**
>
> MEMTO is built upon a vanilla Transformer and stores normal patterns at the time-point level. While this design works well for point anomalies, memory items derived from individual points have a limited ability to capture periodic anomalies.
>
>
> By redesigning the memory module to operate at the patch level, our method offers several advantages.
> - Memory based on patch representations preserves local semantic information more effectively, which improves the detection of periodic anomalies [1].
>
> - Operating on patch-level representations reduces the attention complexity for long sequences in the memory update stage compared with point-level [2].
>
> - This patch-based structure naturally aligns with recent time series foundation models that employ patch tokenization.
>
> **(3) Unified multi-domain training instead of one-model-per-dataset**
>
> Time series foundation models (TSFMs) are pre-trained on diverse datasets and demonstrate strong generalization in forecasting tasks. However, their large capacity can lead to over-generalization in anomaly detection.
>
>
> To address this issue, MOMEMTO augments the TSFM with a patch-based memory module. This design further enables MOMEMTO to operate as a single unified model across multiple domains, which provides several practical benefits:
>
> - It replaces many per-dataset models with one unified model, reducing total training time and GPU memory usage. (Table 4 in the paper)
>
> -  It improves few-shot performance, since domains with limited data can benefit from knowledge shared across other domains. (Figure 2 in the paper)
>
> ***
>
> > **W3.** The paper states that MEMTO follows a “one-model-per-dataset” paradigm. However, the original MEMTO paper mentions that MEMTO can capture multiple prototypes to represent diverse patterns, implying that it could generalize across datasets and potentially achieve a “one-model-for-multiple-datasets” capability. Therefore, this claim in the reviewed paper requires further justification.
>
>
> Although MEMTO can also be trained on multiple domains, its scratch-trained encoder and point-level memory may be less stable across heterogeneous datasets. The table below summarizes how MEMTO, MOMENT, and MOMEMTO perform under different training strategies, allowing a direct comparison across models.
>
> | Model       | AUC-PR | AUC-ROC | VUS-PR | VUS-ROC |
> |------------|:------:|:-------:|:-----:|:-------:|
> | memto      | 6.90   | 52.05   | 11.30 | 56.30   |
> | memto$_{md}$   | 8.73   | 45.69   | 11.87 | 52.13   |
> | moment     | 29.45  | 69.97   | 29.73 | 76.84   |
> | moment$_{md}$  | 29.58  | 69.33   | 29.89 | 76.82   |
> | momento    | 32.83  | 70.00   | 33.23 | 77.49   |
> | momento$_{md}$ | 36.35  | 74.83   | 37.62 | 80.95   |
>
>
>
> [1] Shen, K. Learn hybrid prototypes for multivariate time series anomaly detection.\
> [2] Nie, Y. et al. A Time Series is Worth 64 Words: Long-term Forecasting with Transformers.

---

> > ### Comment · Reviewer_5wBe · 2025-11-26
> >
> > Thank you for your response. The authors' responses to Q1 and W3 have successfully dealt with my concerns of W2, W3 and W4.
> > As for W1, I still think just replacing the encoder with TSFM and replacing the point-wise memory with patch-wise memory is a marginal contribution and lacks novelty.
> > Thus, I will keep my score.

---

> ### Author Response · Authors · 2025-11-26
>
> We are glad to hear that several of your earlier concerns have been addressed, and we agree that both models reconstruct the input through a linear combination of prototypes. With your comment in mind, we would like to outline two key structural differences from MEMTO.
>
> ***
> >**W1.** Compared with MEMTO, the contributions and innovations of this paper are rather marginal. Both approaches build a prototype memory and reconstruct the original data through linear combinations of vectors within that memory. Overall, the work lacks novelty and makes limited contributions to the advancement of the field.
>
>
> **1. Update dynamics**\
>  In MEMTO, all prototypes are updated for every time point of the input sequence.
>
>
>  In our method, only the top-K most relevant prototypes are updated. This selective update strategy avoids unnecessary updates and allows the memory to evolve in a more stable and efficient manner.
>
> **2. The meaning and structure of a prototype**
>
>
>  A single prototype in MEMTO represents a normal pattern learned at the time point level, and each time point in the input is reconstructed using a linear combination that involves all prototypes.
>
>
>  In contrast, a prototype in our method consists of multiple patch-wise sub-prototypes that correspond to the number of patches in the input sequence. Each sub-prototype is aligned with a specific patch position, which preserves the ordering and local temporal structure of the sequence.
>
> During the update stage, refinement is performed at this patch level through attention. This mechanism maintains relationships across patches within each prototype. When an input sequence contains fewer patches than the maximum patch count, the model retrieves and updates only the corresponding subset of sub-prototypes, which allows the memory to adapt flexibly to varying sequence lengths.
>
> Although the final prototype is formed by a linear combination of the top-K selected prototypes, the sub-prototypes corresponding to each patch are linearly combined in a position-aligned manner. This ordered structure enables the memory module to reflect the underlying patterns present in the corresponding parts of the sequence, which helps reduce over-generalization.
> ***
> Once again, thank you for raising this concern. We understand that simply combining a TSFM encoder with a patch-wise memory module may appear to lack novelty. We hope the explanation above clarifies why our design is not a simple substitution but a structural adaptation that enables patch-aligned updates, multi-domain training, and stable integration with patch-based TSFMs.

---

### Official Review · Reviewer_cFPW · 2025-10-29

**Soundness:** 3
**Presentation:** 4
**Contribution:** 3
**Rating:** 6
**Confidence:** 4

**Summary:**

This paper proposes MOMEMTO, a Time Series Foundation Model (TFM) for reconstruction-based time series anomaly detection. The primary goals are to mitigate the "over-generalization" problem, where models accurately reconstruct anomalies, and to address the high computational cost of the "one-model-per-dataset" training paradigm.

Key Contributions:
- Architecture: MOMEMTO integrates a pre-trained TFM encoder from MOMENT with a patch-based memory module made by modifying MEMTO. This module stores prototypes of normal patterns at the patch level to prevent the reconstruction of anomalies.
- Training Strategy: The paper uses a multi-domain training strategy to jointly fine-tune a single model across multiple datasets, which is designed to be more efficient than training separate models.
- Empirical Results: The model is evaluated on 23 univariate benchmark datasets. As a single model, MOMEMTO achieves higher scores on AUC and VUS metrics compared to 13 baseline methods.
- Few-Shot Learning: The method is shown to enhance the performance of its backbone TFM, particularly in few-shot learning scenarios.

On initial read this paper seems very convincing, however a lack of empirical analysis on results and conclusion that contradict reported metrics leave me unconvinced. I want to give this paper a higher rating if these concerns are addressed.

**Strengths:**

Originality: Fair

Quality: Good

Clarity: Good

Significance: Good

Additional note: The paper highlights the key drawbacks of existing methodologies, and modifies MEMTO and MOMENT into one architecture to alleviate drawbacks of both models. The paper is well written and mostly, the motivation for design decisions is well motivated.

**Weaknesses:**

Discrepancy in results and conclusions: In section 4.2 where Figure 3 is discussed, the conclusion is that MOMEMTO exhibits better separation between anomalous and non anomalous scores as compared to MOMENT. However, the results in Appendix C.4 show little or no difference for certain datasets between the two models, and in certain datasets, (TAO-Environment, YAHOO-Synthetic, SMD-Facility, WSD-Webservices), MOMENT outperforms MOMEMTO. I see this as the biggest issue in the paper.

In appendix C.1 in table 9, out of 32 domains, MOMEMTO md has best or 2nd best performance in only 12.
Results: There was no empirical evaluation of the distribution of anomalous scores done, conclusion is made based on (what it seems to be) visual inspection of boxplot graphs.

Uni-variate focus: Only uni–variate datasets were considered in the experiments

**Questions:**

Please provide empirical analysis of results to accompany graphs.

The number of memory items is set to 32, matching the 32 domains identified in the benchmark38. While the update strategy is analyzed (Table 3)39, the paper does not provide a sensitivity analysis for the number of memory items. How would the performance differ if the the number of items < the number of domains. Would the memory items exhibit plasticity? Some experiments with accompanying analysis would be good.

Please provide an explanation for the lack of convincing results in Appendix C.4 and C.1 (Table 9).
A clarification on the ultimate objective of the paper would be appreciated. Is your claim that MOMEMTO is better than all baseline models mentioned? Or that it is a modification that improves MOMENT.

---

> ### Author Response · Authors · 2025-11-21
>
> Thank you for your positive feedback and recognition of our work. We appreciate your valuable comments and would like to address your concerns.
> ***
> > **Q1.** Please provide empirical analysis of results to accompany graphs.
>
>
> To provide a distribution-level comparison (Appendix C.4), we computed the two-sample Kolmogorov–Smirnov (KS) statistic for each domain. The KS statistic measures the maximum difference between the cumulative distributions of normal and anomalous scores, which reflects the degree of separation between the two score distributions. A larger KS value indicates a clearer separation between normal and anomalous scores.
>
>
> 1) **Overall evaluation**
>
> | Metric | MOMENT | MOMENTO |
> |----------|:------:|:-------:|
> | KS       | 0.4270 | 0.5343  |
>
>
> 2) **Per-domain evaluation**
>
> | Dataset     | Sub-domain         | MOMENT | MOMEMTO |Improvement |
> |------------|----------------|------------:|-----------:|--------------:|
> | OPPORTUNITY  | HumanActivity   | 0.277 | 0.911 | 0.634 |
> | NAB         | Environment     | 0.050 | 0.484 | 0.433 |
> | Power       | Facility        | 0.062 | 0.396 | 0.333 |
> | Exathlon    | Facility        | 0.607 | 0.888 | 0.281 |
> | UCR         | Environment     | 0.325 | 0.530 | 0.205 |
> | NEK         | WebService      | 0.477 | 0.665 | 0.188 |
> | SED         | Medical         | 0.128 | 0.311 | 0.183 |
> | UCR         | Sensor          | 0.263 | 0.424 | 0.161 |
> | Stock       | Finance         | 0.519 | 0.652 | 0.133 |
> | UCR         | Medical         | 0.250 | 0.379 | 0.128 |
> | UCR         | Facility        | 0.191 | 0.312 | 0.120 |
> | SVDB        | Medical         | 0.156 | 0.275 | 0.120 |
> | NAB         | WebService      | 0.153 | 0.272 | 0.119 |
> | NAB         | Facility        | 0.226 | 0.331 | 0.105 |
> | UCR         | HumanActivity   | 0.283 | 0.378 | 0.095 |
> | MITDB       | Medical         | 0.109 | 0.201 | 0.092 |
> | LTDB        | Medical         | 0.087 | 0.173 | 0.087 |
> | YAHOO       | WebService      | 0.730 | 0.814 | 0.084 |
> | MGAB        | Synthetic       | 0.105 | 0.184 | 0.079 |
> | IOPS        | WebService      | 0.418 | 0.493 | 0.075 |
> | NAB         | Synthetic       | 0.184 | 0.250 | 0.066 |
> | CATSv2      | Sensor          | 0.280 | 0.339 | 0.060 |
> | SWaT        | Sensor          | 0.616 | 0.655 | 0.039 |
> | TAO         | Environment     | 0.551 | 0.554 | 0.003 |
> | TODS        | Synthetic       | 0.429 | 0.413 | -0.015 |
> | NAB         | Traffic         | 0.131 | 0.107 | -0.023 |
> | YAHOO       | Synthetic       | 0.942 | 0.917 | -0.025 |
> | SMAP        | Sensor          | 0.586 | 0.553 | -0.033 |
> | WSD         | WebService      | 0.500 | 0.437 | -0.062 |
> | MSL         | Sensor          | 0.441 | 0.351 | -0.090 |
> | SMD         | Facility        | 0.628 | 0.501 | -0.127 |
> | Daphnet     | HumanActivity   | 0.699 | 0.103 | -0.596 |
>
> *(Minor differences from the paper may occur because the experiments were rerun, while the overall conclusions stay consistent.)*
>
> For the overall benchmark, our proposed method achieves an average KS value of 53.4, compared to 42.7 for MOMENT. However, at the domain level, MOMEMTO outperforms MOMENT in 24 out of 32 domains, while showing lower values in the remaining 8 domains.
>
> > **Q3.** Please provide an explanation for the lack of convincing results in Appendix C.4 and C.1 (Table 9). A clarification on the ultimate objective of the paper would be appreciated. Is your claim that MOMEMTO is better than all baseline models mentioned? Or that it is a modification that improves MOMENT.
>
> We agree with your comment. The results presented above, along with those in Appendix C.1 and Appendix C.4, are not intended to claim that our method is universally superior across all domains or against all baselines. We included them because they are directly relevant to the rationale behind our experimental design.
>
>
> Conventional benchmarks for time series anomaly detection cover only a limited portion of the heterogeneity observed in real-world time series. Furthermore, threshold-dependent metrics—particularly those based on the best-performing threshold—essentially reflect the upper-bound performance of a method, even though selecting a reliable threshold in practice remains a significant challenge.
>
>
> For these reasons, we adopt a testbed that contains a wide range of heterogeneous time series and employ threshold-independent metrics. Given such heterogeneity, and to the best of our knowledge, no single model has been shown to consistently achieve superior performance across all domains [1].
>
>
> Within this perspective, our proposed method should be viewed as an improved variant of MOMENT for time series anomaly detection. It is not a complete upgrade, but we believe it offers a practically useful improvement tailored for this task.
>
>
> [1] S. Schmidl, P. Wenig, and T. Papenbrock. Anomaly detection in time series: a comprehensive evaluation. Proc. VLDB Endow., 15(9):1779–1797, 2022.

---

> ### Author Response · Authors · 2025-11-21
>
> ***
> > **Q2.** Sensitivity analysis for the number of memory items.
>
> The experimental results with varying numbers of memories are presented in the tables below. In this experiment, we initialized each memory using K-means centroid values.
>
> **1.** Main benchmark results
>
>
> |   | 64    | 32    | 16    | 8     | 4     | 2     | 1     |
> |--------:|:-----:|:-----:|:-----:|:-----:|:-----:|:-----:|:-----:|
> | AUC-PR  | 36.11 | 36.05 | 36.26 | 36.07 | 35.97 | 35.69 | 35.81 |
> | AUC-ROC | 74.59 | 74.22 | 74.18 | 74.19 | 74.30 | 73.25 | 73.50 |
>
> Figure 3 in the attached PDF visualizes the scaled domain similarities computed across the entire dataset, both at the beginning and at the end of training. We observe that the model consistently relies on a small subset of domain memories, regardless of whether the total number of memories is increased or decreased.
>
> **2.** Few-shot results (AUC-PR)
>
>
> | ratio | 0.1   | 0.2   | 0.3   | 0.5   | 0.7   | 0.9   |
> |------:|:-----:|:-----:|:-----:|:-----:|:-----:|:-----:|
> | 32    | 31.02 | 32.84 | 33.40 | 35.04 | 35.23 | 35.63 |
> | 16    | 30.98 | 32.83 | 33.46 | 35.01 | 35.36 | 35.55 |
> | 8     | 30.95 | 32.75 | 33.48 | 34.81 | 35.21 | 35.16 |
> | 4     | 30.93 | 32.42 | 33.52 | 34.69 | 35.42 | 35.32 |
> | 2     | 30.85 | 32.80 | 33.52 | 35.08 | 35.67 | 35.65 |
> | 1     | 30.93 | 32.32 | 33.50 | 34.84 | 35.60 | 35.51 |
>
>
> These results indicate that MOMENTO exhibits stable performance with respect to the number of memories, which is consistent with the sensitivity analysis results reported in MEMTO.

---

> > ### Comment · Reviewer_cFPW · 2025-11-26
> > **Response to rebuttal**
> >
> > Thank you for your rebuttals, the additional tables are informative. I appreciate the clarification that your " proposed method should be viewed as an improved variant of MOMENT for time series anomaly detection."
> > I would suggest to edit the abstract and the main submission in the appropriate places to reflect this statement.

---

> > > ### Author Response · Authors · 2025-11-27
> > >
> > > We appreciate your time and your thoughtful engagement in the rebuttal process, and we hope that our responses have addressed your concerns. Following your suggestion, we have updated the abstract and the relevant sections of the main paper to reflect the clarification.

---

### Official Review · Reviewer_JqEL · 2025-11-01

**Soundness:** 3
**Presentation:** 4
**Contribution:** 3
**Rating:** 6
**Confidence:** 3

**Summary:**

This paper introduces MOMEMTO, a memory-augmented anomaly detection framework that integrates the pre-trained MOMENT encoder with the memory update mechanism of MEMTO.
The model is designed for multi-domain, univariate time series and operates in a fully data-driven manner—learning cross-domain normality representations without domain labels.
Key components include a patch-level memory alignment strategy that adaptively refines prototypical memory items across heterogeneous domains.
Empirically, MOMEMTO achieves superior performance over strong baselines (MOMENT, MEMTO, TimesNet, Anomaly Transformer) on the TSB-AD-U benchmark while maintaining computational efficiency.

**Strengths:**

**Originality:**
MOMEMTO combine a pre-trained cross-domain encoder with a memory mechanism for anomaly detection. Its data-driven update policy allows unsupervised memory adaptation without explicit domain label

**Quality:**
Good

**Clarity:**
The formulation of memory alignment, gating, and normalization is well structured, and the Appendix provides sufficient hyperparameter details for reproducibility.

**Significance:**
The method simultaneously improves multi-domain generalization and computational efficiency—achieving better accuracy with fewer active memory items. This is practically important for scalable time-series anomaly detection in resource-constrained or heterogeneous environments.

**Weaknesses:**

**1. Limited scope of experimental validation:**
While the authors acknowledge this limitation in their conclusion, the empirical validation remains confined to univariate time series from a single benchmark (TSB-AD-U), which constrains the generalizability of the results. Additional experiments on either multivariate datasets or alternative univariate benchmarks would be essential to substantiate the robustness and broader applicability of the proposed framework.

**2. Interpretability of Figure 4:**
The description of “accumulated during training” remains ambiguous—it’s unclear whether the visualization reflects cumulative statistics over the entire training process or a specific epoch window.
Clarifying this temporal scope would make the observed concentration dynamics more interpretable.

**3. Domain-level selection imbalance:**
In Figure 4, the Daphnet (Human Activity) domain exhibits the strongest memory selection, while some domains appear nearly unused.
The paper does not analyze why this imbalance occurs—whether due to domain size, feature diversity, or intrinsic signal variance.
Additional quantitative analysis or ablation would strengthen the claim of cross-domain adaptability.

**Questions:**

**1. Comparison with H-PAD:**
The proposed framework shares conceptual similarities with H-PAD (Hybrid Prototype-based Anomaly Detection), which also employs patch-level prototypes but introduces periodic prototypes for long-range temporal modeling.
Could the authors elaborate on the principal design differences—especially regarding prototype granularity and update dynamics—and whether H-PAD could serve as a complementary baseline?

**2. Domain-level bias in memory utilization:**
What drives the dominant selection of the Daphnet :Human Activity domain in Figure 4?
Conversely, which domains are rarely or never selected, and do these cases correlate with domain size or anomaly density?
A systematic analysis would clarify whether the model’s “data-driven” selection is influenced by statistical imbalance.

**3.Definition of “accumulated during training”:**
Does Figure 4 aggregate similarities across all epochs or represent a snapshot at a particular stage of training (e.g., final epoch)?
A precise definition is necessary to interpret whether the observed concentration reflects progressive convergence or a steady-state behavior of the memory update dynamics.

---

> ### Author Response · Authors · 2025-11-21
>
> Thank you for your thoughtful and constructive comments. We appreciate the time and effort you invested in reviewing our work.
> ***
> > **Q1.** Comparison with H-PAD
>
>
> We agree that H-PAD and our framework share conceptual similarities in adopting patch-level memory to mitigate over-generalization. Below, we clarify the principal design differences, focusing on prototype granularity and update dynamics.
>
>
> **1) Prototype granularity**
>
> H-PAD learns two distinct types of prototypes.
>
>
> (i) Multi-scale patch prototypes
>
>
> H-PAD constructs patch representations by first applying average pooling over neighboring time points for multi-scale patch prototypes. Based on these pooled representations, it builds separate prototype sets for different patch lengths.
>
> (ii) Period prototypes
>
>
> Period prototypes are derived by detecting dominant frequencies through FFT and constructing, for each channel, *a prototype that summarizes the dominant period-aligned segments.* This allows each channel’s characteristic periodic behavior to be captured by the prototype.
>
>
> Overall, this hybrid formulation is designed to model both local interval anomalies and longer-range periodic anomalies through dedicated prototype groups, providing a fine-grained and structurally rich prototype space.
>
> In contrast, MOMEMTO uses a single type of prototype.
> Our prototypes are fixed-length patch-level representations directly aligned with the patch embeddings of the MOMENT encoder. These prototypes capture normal patterns across domains, and each prototype preserves the original ordering of patches because the prototype layout follows the positional structure of the input window.
>
> This design preserves both the information within each patch and the relationships across patches, and *the ordered structure of the prototype can reflect the underlying patterns present in the corresponding part of the sequence.*
>
> In summary, H-PAD provides a highly granular prototype structure that naturally aligns with multivariate settings. Our method adopts a simpler yet effective design that emphasizes preserving and leveraging patch-level information.
>
> **2) Update dynamics**\
> Both H-PAD and MOMEMTO employ the gated update mechanism introduced in MEMTO. The key difference lies in how the update targets are determined.
>
>
> H-PAD performs broad updates within each branch. Once the multi-scale patch representations or the period-aligned segments are computed, all prototypes associated with that scale or period participate in the update. The model updates the entire prototype group using the corresponding embeddings, allowing it to aggregate information across all patches or all period segments for each channel.
>
>
> In contrast, MOMEMTO updates prototypes selectively. For each input sequence, the model retrieves prototypes based on the sequence length and identifies only the top-K prototypes that are most similar to the current query. Patch-level attention is then applied to update only these selected prototypes. This focused update strategy avoids updates to unrelated prototypes and enhances stability and computational efficiency when training a unified model across heterogeneous domains.
>
>
> In summary, H-PAD updates all prototypes within a given group, whereas MOMEMTO refines only the most relevant prototypes for each input. This distinction reflects different design goals: H-PAD aggregates information broadly for multi-scale and periodic modeling, while MOMEMTO prioritizes selective and stable prototype refinement aligned with patch-level representations.
>
>
> We agree that H-PAD could be a valuable complementary baseline. However, we could not include it because we were unable to find an official or publicly available implementation.

---

> ### Author Response · Authors · 2025-11-21
>
> ***
> > **Q2.** Domain-level bias in memory utilization
>
> Thank you for the insightful comment. We also expected the same behavior when conducting additional experiments. Interestingly, the empirical results revealed a consistent and informative pattern, which we summarize below.
>
>
> First, memory initialization is performed by averaging randomly selected samples from each domain. Since these samples vary for each training run, the memory initialization value and the set of memory items that are primarily updated and utilized also vary.
>
>
> - Fig. 1 in the attached PDF visualizes the number of sub-sequences used from each domain in the training set (x-axis) and the accumulated domain weights throughout the entire training (y-axis). While larger domains occasionally appear in the set of influential memory items, this trend is neither persistent nor dominant.
>
>
> - As shown in Fig. 2 in the attached PDF, we track the ranking of cosine similarity between the full training dataset and each memory item throughout training, evaluated every 50 update steps (snapshot). Memory items that eventually become influential generally exhibit higher similarity at initialization, while less influential ones start and remain at lower similarity levels.
>
>
> In summary, these results suggest that both the representational structure of the pre-trained encoder and the randomness in selecting samples for memory initialization have a meaningful impact on which memory items become dominant. The model tends to reinforce memory items that are naturally more aligned with the overall dataset. At the same time, although the initialization involves random sampling, the model’s data-driven update strategy reliably yields stable performance and consistently reproduces the similarity-based convergence patterns described above across repeated trials. These findings suggest that the convergence of domain memory items is not strongly influenced by domain size.
>
>
> ***
> >**Q3.** Definition of “accumulated during training”
>
>
> Thank you for pointing out the ambiguous phrasing. The term “accumulated during training” refers to the accumulated similarities over the entire training process, rather than a snapshot at a specific epoch. We have updated both the main text and the annotation in Figure 4 for clarity in the revised version.

---

### Official Review · Reviewer_iLFi · 2025-11-01

**Soundness:** 2
**Presentation:** 2
**Contribution:** 2
**Rating:** 4
**Confidence:** 4

**Summary:**

This paper proposes the model MOMEMTO, which is a type of basic model for anomaly detection in time series. It is enhanced through a patch-based memory module by utilizing a pre-trained encoder and patch-based memory module, which somewhat alleviates the overgeneralization problem.

**Strengths:**

1. A more fair metric evaluation system was adopted, reasonably reflecting the model's performance in terms of accuracy;

2. The idea of a model handling datasets from multiple domains has been widely explored in the time series field, but it still holds some value in the TSAD domain;

3. The explanation of the method in the article is relatively clear.

**Weaknesses:**

1. The novelty of the article is relatively limited and needs to be restated, especially with further experimental or theoretical explanations of the mutual benefits among the components. The proposed framework combines three elements: the Patch mechanism, the memory mechanism, and the Moment model. However, these three components have likely been widely explored in the TSAD field, particularly the first two, which leads me to believe that this paper may have difficulty providing readers with novel insights.

2. The paper claims to be a TFM specifically designed for time series anomaly detection that supports multiple domains. On one hand, this is positive, as we encourage more research in this area adapting a single model to multiple time series anomaly datasets. On the other hand, according to experimental comparison results, machine learning models like Isolation Forest and deep learning models like LSTM-AD show nearly suboptimal performance. Although they do not achieve the performance level of existing models, they can reach performance close to that of the proposed model with a simpler structure and very low cost. In contrast, the TFM dedicated to time series anomaly detection may incur a much higher cost without showing remarkable performance, and to some extent, it loses the ability to handle other time series tasks. Therefore, I believe the motivation of this work should be re-evaluated. Additionally, I think comparisons with more integrated machine learning models should be conducted to assess whether higher costs are justified for relatively modest performance gains. More experiments and discussions are needed to support the rationale behind this model.

3. The comparative baselines in the article are relatively weak, with few methods from 2025, especially in the deep learning section where many methods are outdated. This makes the comparative experiments somewhat unreliable and further calls into question whether the motivation of the paper is reasonable or robust. It is recommended that the authors supplement more comparative baselines.

4. The discussion on computational efficiency is insufficient and too vague, and this indicator of other comparative algorithms should be included.

**Questions:**

1. The content regarding the novelty of the model and the rationality of combining existing technologies needs more theoretical and experimental evidence. It is recommended that the author provide additional support;

2. The comparison methods are too outdated;

3. The advantages of the model are not obvious, especially more comprehensive comparative experiments on parameters, training, and inference time are needed to highlight the advantages of the proposed method.

---

> ### Author Response · Authors · 2025-11-21
>
> Thank you for dedicating your valuable time to reviewing our paper. Your insightful comments have helped us reconsider important aspects of our research.
> ***
> > **Q1.** The content regarding the novelty of the model and the rationality of combining existing technologies ... recommended that the author provide additional support.
>
> Recent advances in time series deep models have been driven by the adoption of the patch mechanism, beginning notably with PatchTST [1]. Since then, patch-level tokenization has become a common design choice in many models, and the effectiveness of masked patch-based self-supervised learning has further accelerated progress in time series foundation models (TSFM). With large-scale pretraining and increased model capacity, recent TSFMs demonstrate strong generalization ability, achieving competitive zero- and few-shot performance in time series forecasting[2, 3].
>
>
> However, these benefits do not directly extend to time series anomaly detection (TSAD). __(i) Increasing model size and representational capacity can lead to over-generalization, where unseen anomalous patterns are also reconstructed accurately.__ In 2023, MEMTO [4] introduced the first memory architecture for TSAD, showing the effectiveness of memory mechanisms. However, __(ii) it still relied on time point-level tokenization, which is not fully aligned with the patch-based architectures used in recent TSFMs.__  As a result, __(iii) developing a single unified TSAD model remains challenging__ compared to forecasting tasks.
>
>
> Motivated by these considerations, our model design is based on the following rationale.
>
>
> 1. **Why MOMENT with Patch mechanism?**
>    - Although most TSFMs are primarily designed for forecasting, MOMENT supports several downstream tasks by simply changing the task-specific head.
>    - MOMENT’s patch-based encoder provides well-structured time series representations, as patch tokenization preserves local semantic information more effectively than point tokenization [1, 3].
>
>
> 2. **Why Patch mechanism with Memory mechanism?**
>    - Memory mechanisms built on point-level representations are mainly effective for detecting point anomalies, as they learn prototypes at the individual time point, making interval and periodic anomalies more difficult to capture [5].
>    - A patch-based memory architecture inherits the advantages of the patch mechanism and offers structural compatibility with recent patch-based architectures.
>
> 3. **Why MOMENT with Memory mechanism?**
>    - While MOMENT offers strong representational capacity, it still inherits the over-generalization issue commonly observed in reconstruction-based TSAD models. Memory mechanism helps mitigate this limitation by guiding reconstruction toward normal patterns and reducing the tendency to accurately reconstruct unseen anomalies.
>    - Memory mechanisms are sensitive to initialization [4], but this challenge can be alleviated by leveraging the well-organized embedding space provided by MOMENT’s pretrained encoder. (Chapter 4.2 Table 2)
>
> ***
>
> **(i) Over-generalization**
>
> As shown in Table 1 of Chapter 4.1 and in the additional experimental tables below (Q2), the reconstruction-based TSFM MOMENT does not exhibit a large performance gap compared with existing models, despite its substantial model size.
>
> **(ii) Patch mechanism**\
> As shown in the table below, when the three models are compared with each other, applying the patch mechanism to both the tokenization and the memory module empirically improves anomaly detection performance.
>
> | Method      | AUC-PR | AUC-ROC | VUS-PR | VUS-ROC |
> |------------|:------:|:-------:|:-----:|:-------:|
> | memto      | 6.90   | 52.05   | 11.30 | 56.30   |
> | memto$_{md}$   | 8.73   | 45.69   | 11.87 | 52.13   |
> | moment     | 29.45  | 69.97   | 29.73 | 76.84   |
> | moment$_{md}$  | 29.58  | 69.33   | 29.89 | 76.82   |
> | momemto    | 32.83  | 70.00   | 33.23 | 77.49   |
> | momemto$_{md}$ | 36.35  | 74.83   | 37.62 | 80.95   |
>
> **(iii) A unified TSFM for TSAD**\
> Both MEMTO and MOMENT show only limited performance gains when trained in a multi-domain setting as a single model. In contrast, MOMEMTO achieves improved performance as a unified model, and the few-shot experiment in Chapter 4.1(Figure 2) further demonstrates its practicality. The consistent performance improvement as the proportion of training data increases empirically supports its effectiveness as an integrated model.
>
>
>
>
> We believe that these considerations help clarify and support the architectural decisions in our model.
>
>
>
> [1] Nie, Y. et al. A Time Series is Worth 64 Words: Long-term Forecasting with Transformers.\
> [2] Liu, Y. et al. Sundial: A Family of Highly Capable Time Series Foundation Models.\
> [3] Goswami, M. et al. MOMENT: A Family of Open Time-series Foundation Models.\
> [4] Song, J. et al. MEMTO: Memory-guided Transformer for Multivariate Time Series Anomaly Detection.\
> [5] Shen, K. Learn hybrid prototypes for multivariate time series anomaly detection.

---

> ### Author Response · Authors · 2025-11-21
>
> ***
> >**Q2.** The comparison methods are too outdated.
>
> We conducted additional experiments using the recent time series anomaly detection models DADA [1] and KAN-AD [2], and the results are presented in the table below.
>
>
>
> [1] Shentu, Q. et al. Towards a General Time Series Anomaly Detector with Adaptive Bottlenecks and Dual Adversarial Decoders. ICLR, 2025\
> [2] Zhou, Q. et al. KAN-AD: Time Series Anomaly Detection with Kolmogorov–Arnold Networks. ICML, 2025
>
>
>
> | Model               | AUC-PR | AUC-ROC | VUS-PR | VUS-ROC | Train Time | Infer Time | Model Size |
> |--------------------|:------:|:-------:|:------:|:-------:|:----------:|:----------:|:----------:|
> | LOF                | 14.24  | 58.17   | 16.87  | 67.74   | 1h 45m 46s | 1h 49m 44s | -          |
> | OCSVM              | 15.85  | 65.13   | 22.51  | 73.49   | 7h 7m 56s  | 16h 40m 3s | -          |
> | Isolation Forest   | 29.04  | $\underline{71.18}$ | 29.80 | 77.51   | 12m 23s    | 2m 53s     | -          |
> | MEMTO              | 6.90   | 52.05   | 11.30  | 56.30   | 47m 4s     | 13s        | 2B         |
> | Donut              | 15.28  | 55.97   | 20.78  | 68.56   | 4h 27m 48s | 17m 8s     | 7.25M      |
> | TimesNet           | 18.10  | 61.19   | 25.91  | 72.03   | 3h 51m 19s | 21m        | 25.7M      |
> | AT                 | 19.02  | 62.53   | 23.61  | 73.05   | 2h 17m 4s  | 17m 21s    | 16.6B      |
> | TranAD             | 24.67  | 63.66   | 29.95  | 73.87   | 4h 55m 11s | 11m 54s    | 0.13M      |
> | OmniAnomaly        | 29.87  | 69.45   | 32.15  | 76.49   | 1h 54m 20s | 7m 23s     | 4.84M      |
> | LSTM-AD            | 31.77 | 67.70 | 32.66 | 75.74 | 3h 37m 21s | 17m 52s   | 3.64M     |
> | **KAN-AD**             | 29.20  | 69.54   | 30.25  | 76.30   | 3h 18m 48s | 3m 51s     | 0.1M       |
> | **DADA**               | $\underline{33.59}$ | 70.42 | $\underline{34.06}$ | $\underline{77.62}$ | 14s | 10m 35s | 64.5M |
> | Chronos-base       | 26.43  | 65.64   | 27.18  | 72.91   | 1h 11m 26s | 7h 51m 13s | 70B        |
> | TimesFM            | 28.43  | 67.29   | 29.88  | 74.20   | 11m 36s    | 2h 54m 40s | 70B        |
> | MOMENT             | 29.45  | 69.97   | 29.73  | 76.84   | 23m 23s    | 54s        | 134.75B    |
> | MOMEMTO   | 32.83  | 70.00   | 33.23  | 77.49   | 24m 7s     | 1m 20s     | 135.8B     |
> | MOMEMTO$_{md}$     | **36.35**  | **74.83** | **37.62** | **80.95** | 1m 27s | 54s | 388M |
>
>
>
> >**Q3**. The advantages of the model are not obvious, especially more comprehensive comparative experiments on parameters, training, and inference time are needed to highlight the advantages of the proposed method.
>
> The above experiments (Q2) were conducted on a server equipped with an Intel i9-10900X CPU and an NVIDIA RTX 4090 GPU, which is consistent with the specification used in the paper.
>
>
> Given the large scale of the train set (2,441,731 points) and test set (18,160,361 points), models that rely on sliding window-based processing may incur substantial computational cost. As a result, training and inference time vary significantly depending on model design, efficiency, and hyperparameter settings. Classical machine learning baselines are executed on the CPU, which further increases runtime in large-scale settings. Forecasting-oriented TSFMs may also incur overhead due to densely overlapping windows and large model capacity.
>
>
> Based on these results, we believe that MOMEMTO can serve as a single unified model that offers practical advantages in model size and helps reduce the need to load and manage multiple separate models.

---

### Author Response · Authors · 2025-11-21

Attachment for rebuttal \
[⬇️ PDF](https://drive.google.com/uc?export=download&id=1G_UcvX8DOi_4tGzbzPwbFqMjNI3Wo8bA)

---

### Meta-Review · Area_Chair_cwG5 · 2026-01-07

**Summary:**

The paper proposes MOMEMTO, a time series foundation model designed for anomaly detection. It leverages the pre-trained encoder from MOMENT and introduces a patch-based memory module to alleviate the over-generalization problem.

While reviewers acknowledged that the overall methodology is explained with relative clarity, there are significant concerns regarding the paper's contribution and novelty. Two reviewers pointed out that the work appears incremental compared to MEMTO, primarily consisting of replacing the encoder with a foundation model and shifting from point-wise to patch-wise memory. More specifically, it is a typical A+B paper, where MOMENT + MEMTO is combined. Furthermore, the baselines selected for comparison are outdated. In addition, placing DADA under the deep learning category is questionable as DADA is one of the first batch of foundation models for time series outlier detection with zero-shot capabilities.

I read the paper and the author's response to the reviewers and believe these issues still exist. Consequently, I recommend rejecting the paper.

**Reviewer Concerns:**

Addressed Concerns:
1.	Computational Efficiency: A discussion and comparison of computational efficiency between MOMEMTO and baselines were provided.
2.	Additional anomaly detection metrics, such as the F1 score, were included.
3.	Memory Initialization: The authors clarified how MOMEMTO addresses the critical issue of memory initialization.

Unresolved Concerns:
1.	Limited Novelty: The distinct contributions and novelty of MOMEMTO remain insufficiently justified compared with MEMTO.
2.	Outdated Baselines: The experimental comparison lacks recent advancements. The authors should include more contemporary baselines (such as: ModernTCN, CATCH, GPT4TS).
3.	Regarding domain-level bias in memory utilization, the authors should design better methods to address the imbalance across different domains, not just random sampling for each domain.

**Reviewer Scores:**

Reviewer 5wBe explicitly stated maintain the original score after the rebuttal.
The remaining three reviewers did not respond during the discussion period. Based on the rebuttal content and the persistence of the major concerns, I think the other reviewers would not increase their initial scores.

---

### Decision · Program_Chairs · 2026-01-26

Reject